# Adjusting the energy of interfacial states in organic photovoltaics for maximum efficiency

Nicola Gasparini [1,2,14 ✉], Franco V. A. Camargo [3,14], Stefan Frühwald[4], Tetsuhiko Nagahara [3,5], Andrej Classen [2], Steffen Roland[6], Andrew Wadsworth [7], Vasilis G. Gregoriou[8,9], Christos L. Chochos [8,10], Dieter Neher [6], Michael Salvador[11], Derya Baran [11], Iain McCulloch [7,11], Andreas Görling[4], Larry Lüer [2✉], Giulio Cerullo [3] & Christoph J. Brabec [2,12,13✉]

A critical bottleneck for improving the performance of organic solar cells (OSC) is minimising non-radiative losses in the interfacial charge-transfer (CT) state via the formation of hybrid energetic states. This requires small energetic offsets often detrimental for high external quantum efficiency (EQE). Here, we obtain OSC with both non-radiative voltage losses (0.24 V) and photocurrent losses (EQE > 80%) simultaneously minimised. The interfacial CT states separate into free carriers with ≈40-ps time constant. We combine device and spectroscopic data to model the thermodynamics of charge separation and extraction, revealing that the relatively high performance of the devices arises from an optimal adjustment of the CT state energy, which determines how the available overall driving force is efficiently used to maximize both exciton splitting and charge separation. The model proposed is universal for donor: acceptor (D:A) with low driving forces and predicts which D:A will benefit from a morphology optimization for highly efficient OSC.

[1] Department of Chemistry and Centre for Plastic Electronics, Imperial College London, London, UK. [2] Institute of Materials for Electronics and Energy Technology (I-MEET), Friedrich Alexander-University Erlangen-Nuremberg, Erlangen, Germany. [3] Dipartimento di Fisica, IFN-CNR, Milano, Italy. [4] Department of Chemistry and Pharmacy, Friedrich Alexander-University Erlangen-Nuremberg, Erlangen, Germany. [5] Department of Chemistry and Materials Technology, Kyoto Institute of Technology, Kyoto, Japan. [6] Institut für Physik und Astronomie Physik weicher Materie University of Potsdam, Potsdam, Germany. [7] Department of Chemistry, Chemistry Research Laboratory, University of Oxford, Oxford, UK. [8] Advent Technologies SA, Patras, Greece. [9] National Hellenic Research Foundation, Athens, Greece. [10] Institute of Chemical Biology, National Hellenic Research Foundation, Athens, Greece. [11] Division of Physical Sciences and Engineering (PSE), KAUST Solar Center (KSC), King Abdullah University of Science and Technology (KAUST), Thuwal, Saudi Arabia. [12] Bavarian Center for Applied Energy Research (ZAE Bayern), Erlangen, Germany. [13] Helmholtz-Institute Erlangen-Nürnberg (HI ERN), Erlangen, Germany. [14] These authors contributed equally: Nicola Gasparini, Franco V.A. Camargo. ✉email: n.gasparini@imperial.ac.uk; larry.lueer@fau.de; christoph.brabec@fau.de

According to the Shockley and Queisser description, an ideal solar cell exhibits only unavoidable radiative losses[1]. Operated as a light-emitting diode, such a device should deliver electroluminescence (EL) external quantum efficiency (EQE$_{EL}$) of unity[2]. For inorganic (GaAs) and hybrid (perovskites) solar cells, EQE$_{EL}$ close to 100% is indeed observed, while in organic photovoltaic (OPV) devices, EQE$_{EL}$ is orders of magnitude lower, indicating the dominance of non-radiative pathways for charge recombination[3–6]. These losses are mainly caused by recombination of the charge transfer (CT) states generated at the interface between the electron donor (D) and acceptor (A) materials[7–11]. Because the energy of the CT state ($E_{CT}$) is lower than the energy ($E_{LE}$) of the localised excitations (LE), the rate of non-radiative recombination is expected to increase following the energy gap law[12–14]. The recent introduction of non-fullerene acceptors (NFAs), with tunable energy levels and optical absorption, greatly improved the power conversion efficiencies (PCE) in OPV, which have now reached 17% in single-junction devices[15–18]. However, the non-radiative open circuit voltage losses ($\Delta V_{OC,nr}$) are still substantial and reduction of $\Delta V_{OC,nr}$ is critical for further improvements[19–21]. The hybridisation of LE and CT states, causing the transfer of oscillator strength towards the weakly allowed CT transition, has been proposed as a means to reduce $\Delta V_{OC,nr}$[22,23]. However, a low CT stabilization energy $\Delta E_{LE,CT} = E_{LE} - E_{CT}$ can lead to incomplete exciton dissociation, thus penalizing the EQE[24,25]. Therefore, a complete physical understanding of the interplay between state hybridization and non-radiative voltage losses is essential for the development of next-generation OPV technologies.

Here, we demonstrate a pathway to minimise non-radiative voltage losses in an OPV device without penalizing EQE. We introduce a bulk heterojunction (BHJ) solar cell based on polymer:NFA[26] blends that combine relatively high power conversion efficiency (PCE > 12%) with high EQE$_{EL}$ ($10^{-4}$) and low non-radiative loss $\Delta V_{OC,nr} = 0.24$ V. We employ femtosecond transient absorption (TA) spectroscopy, coupled to a global analysis technique, to unravel the kinetics of electron and hole transfer, in addition to charge separation, in the blend. We gain further insight into the electronic structure of the interfacial CT state through quantum chemical calculations, which we apply to model the population equilibrium between interfacial and bulk states from EL and EQE spectra. We show that this equilibrium is decisively controlled by the degeneracy and energy of the interfacial CT states and that it attains a near-optimum value in WF3:O-IDTBR, explaining its ability to produce a high open-circuit voltage ($V_{OC}$) and EQE at the same time. We generalise these results by showing that, for any blend with a low driving force, there exists an optimum CT energy, which is at the crossing point of normalised $V_{OC}$ and EQE curves along the CT energy axis. We discuss known methods to adjust the CT energy in a given blend in light of this new functional relationship.

## Results

### Photovoltaic characterisations and non-radiative voltage losses.
The medium bandgap D–A copolymer WF3 was selected as electron donor material, while the electron acceptor was either the fullerene derivative PC$_{70}$BM or the small molecule NFA O-IDTBR (Fig. 1a). The optical bandgap of neat WF3 is 1.85 eV and this value is not affected when blended with PC$_{70}$BM. The optical bandgap of the WF3:O-IDTBR blend is 1.6 eV (Fig. 1b), which lies between the optical bandgaps of neat O-IDTBR annealed and as-deposited films (1.58 and 1.65 eV, respectively, see Supplementary Fig. 2). This shows that the absorption onset of WF3:O-IDTBR blends can be assigned to small aggregates of O-IDTBR in

contact with WF3; the absorption spectra show no evidence of an interfacial CT state (Supplementary Fig. 3).

Figure 1c shows the current-voltage characteristics measured at 1 sun condition for the OSCs (in an inverted configuration, for details see Supplementary Information 9). WF3:PC$_{70}$BM and WF3:O-IDTBR based devices delivered a PCE of 9% and 12%, respectively, the latter showing a high open-circuit voltage $V_{OC} = 1.06$ V and high EQE over 80% in the 1.6–1.8 eV region (Fig. 1d). Compared to the optical bandgap $E_g$, calculated by the intersection between the absorption and PL spectra, the total $V_{OC}$ loss in the WF3:O-IDTBR device is only $\Delta V_{OC,tot} = 1/q \cdot E_g - V_{OC} = 0.56$ V, where $q$ is the elementary charge. The total $V_{OC}$ losses can be thought of as the sum of unavoidable radiative and non-radiative losses: $\Delta V_{OC,tot} = \Delta V_{OC,rad} + \Delta V_{OC,nr}$. The latter can be obtained from EQE$_{EL}$ via $\Delta V_{OC,nr} = -k_B T/q \cdot \ln(\text{EQE}_{EL})$, where $k_B$ is Boltzmann's constant and $T$ is the cell's operational temperature[13,27]. The EQE$_{EL}$ values of WF3:O-IDTBR and WF3:PC$_{70}$BM devices are $10^{-4}$ and $10^{-6}$, corresponding to $\Delta V_{OC,nr}$ of 0.24 V and 0.34 V, respectively (Fig. 1g for details see Supplementary Section I). The EL of the WF3:O-IDTBR blend is clearly visible by the naked eye (inset of Fig. 1f) even though the emission wavelength approaches the long-wavelength limit of human vision. The EL spectrum of the WF3:O-IDTBR blend matches that of the pure O-IDTBR, which suggests that $E_{LE}$ and $E_{CT}$ have similar values and thus $\Delta E_{LE,CT}$ is small. A non-radiative voltage loss of 0.24 V, as obtained for WF3:O-IDTBR, is among the lowest values ever reported for OPV blends[27–29].

Non-radiative voltage losses are strongly reduced when $E_{LE}$ and $E_{CT}$ are very similar[22,29]. However, different from Ref. 22. where the blends with the lowest $\Delta V_{OC,nr}$ suffered from low fill factors (FF) and short-circuit current densities ($J_{SC}$), resulting in low EQE, the WF3:O-IDTBR blend retains high EQE in combination with its low $\Delta V_{OC,nr}$. In the aforementioned work[22], the photocurrent losses have been attributed to the increased total recombination rate due to a higher oscillator strength for the CT transition, caused by hybridization of the LE and CT states. In order to obtain design principles towards maximum $V_{OC}$ in OPV, it is therefore important to understand the reason for the observed combination of low non-radiative losses and high EQE in the WF3:O-IDTBR blends.

### Ultra-fast spectroscopy analyses of WF3:O-IDTBR devices.
To get insight into the dynamics of the CT and separation processes, we use femtosecond TA spectroscopy. The samples are excited by a 10-fs pulse, mainly in resonance with WF3 excitons (shaded grey in Fig. 2C), and probed with a broadband continuum covering the 1.3–2.2 eV photon energy range. We choose broadband excitation because the resulting 10-fs pulses yield a much better time resolution compared to standard TA measurements with a time resolution of the order of 100–200 fs, allowing us to observe photophysical processes from their very onset. Figure 2C shows TA spectra at 20 ps pump-probe delay, following excitation with a 10 fs pulse (shaded grey), of films of neat WF3 (blue dots) and annealed O-IDTBR (green dots), along with the fits obtained from the global analysis (solid lines, see Supplementary Section 2 for modelling details) and the normalised ground state absorption, shown in dotted lines. The TA of WF3 includes a photobleaching (PB) signal peaking at 1.87 eV, corresponding to the bandgap absorption and its vibronic progression, a broad photoinduced absorption (PA) band in the near-infrared (NIR), and no stimulated emission (SE). These signatures are due to the formation of a dark interchain exciton (also known as a polaron pair) within 1 ps (see Supplementary Section 2), which being an

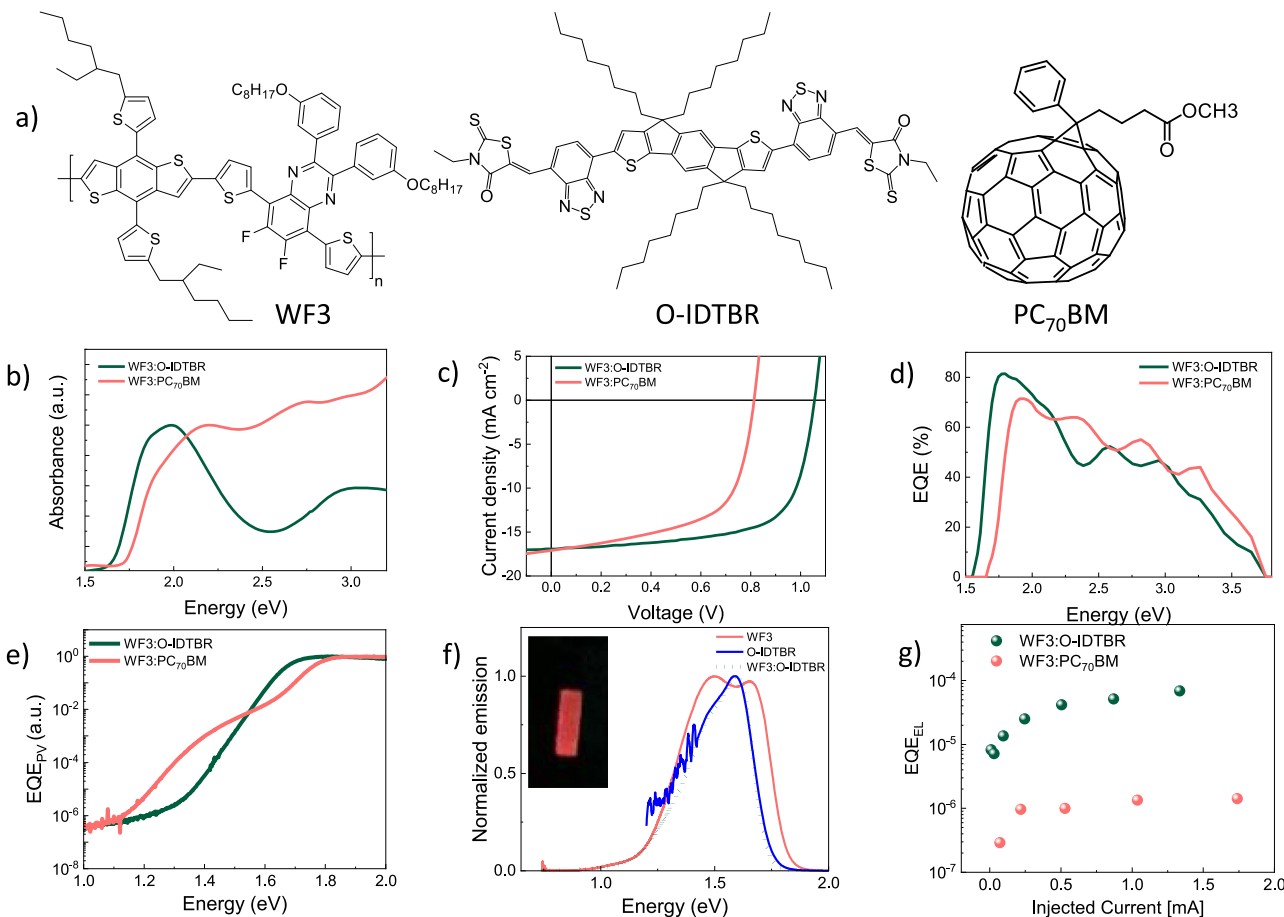

**Fig. 1 Steady-state characterisation of organic solar cells. a** Chemical structures of the donor and acceptor materials used in this work. **b** Absorption spectra of WF3:O-IDTBR and WF3:PC$_{70}$BM blends. **c** Current density–voltage characteristics under 1 sun illumination for WF3:O-IDTBR and WF3:PC$_{70}$BM devices. **d** External quantum efficiency measurements of the same devices as shown in (**c**). **e** Sensitive EQE measurements (EQE$_{PV}$) for the same devices as shown in (**c**). **f** Electroluminescence spectra of the pristine materials and the WF3:O-IDTBR blend. The inset depicts a photograph of the WF3:O-IDTBR electroluminescence (the original photograph can be found in the SI). **g** EQE$_{EL}$ of WF3:O-IDTBR and WF3:PC$_{70}$BM devices as a function of the driving current.

excited state retains PB but suppresses SE, so that the weak steady-state emission from WF3 films can be explained by the Boltzmann equilibrium between bright singlet excitons and the less energetic dark interchain excitons (see Supplementary Fig. 5). The TA spectrum of annealed neat O-IDTBR films, on the other hand, contains a PA band in the NIR, a PB corresponding to the vibronic progression from the absorption spectrum (at 1.68 and 1.9 eV) along with its mirror image SE peak at 1.4 eV, thus corresponding to singlet excitons. The (00) vibronic transitions of the SE and PB bands are nearly completely superposed. Spectral decomposition (Supplementary Table 1) yields a Stokes shift of only about 80 meV, showing that there is very little geometric relaxation along with low energy torsional modes.

To discuss the TA of the blend, we refer to Fig. 2A, where we schematically represent an area around the D/A interface and the possible photoinduced states, and Fig. 2B, showing the corresponding simplified cartoons of the TA signatures. Besides the LEs from WF3 and O-IDTBR (shown in blue and green, respectively), charged states are possible both at the interface (CT, shown in gold) or when at least one charge has diffused away from the interface (charge-separated states, CS, also shown in gold). Both CT and CS states are represented by the same colour because they present very similar intrinsic transient signatures, including PB of both charged organic semiconductors as well as a PA band in the NIR, whose spectral sensitivity to the

environment (bulk or interface) we cannot probe due to spectral limitations. However, they can be distinguished by consideration of the Stark effect that they introduce upon neighbouring molecules by the electric field between positive and negative charges: as shown in Fig. 2A, the strongest Stark effect is experienced if there is a molecule between the positive and negative charge, which is the case only for a CS state but not for a CT state (for a simple electrostatic calculation, see Supplementary Section 5). These molecules will then produce a TA signal shaped as the derivative of their ground state absorption, which is often referred to as "electroabsorption" (EA)[30].

Figure 2D shows the TA spectrum of the WF3:O-IDTBR blend 50 fs after excitation with 10 fs broadband pulses predominantly in resonance with the WF3 exciton (black dots). Pump and probe beams have parallel polarization; we also performed the study at magic angle polarization, qualitatively showing similar results. For a quantitative evaluation, we preferred parallel polarization due to the higher signal quality.

Derivative-shaped EA features of both WF3 and O-IDTBR are already clearly present, resulting in a spectrum that is very different from the one that would result from the superposition of the TA spectra from the pure materials in Fig. 2C. Hence, CT after WF3 excitation is extremely rapid and not diffusion-limited, outcompeting the formation of polaron pairs seen in the pristine WF3 films. This can be associated with high excess energy

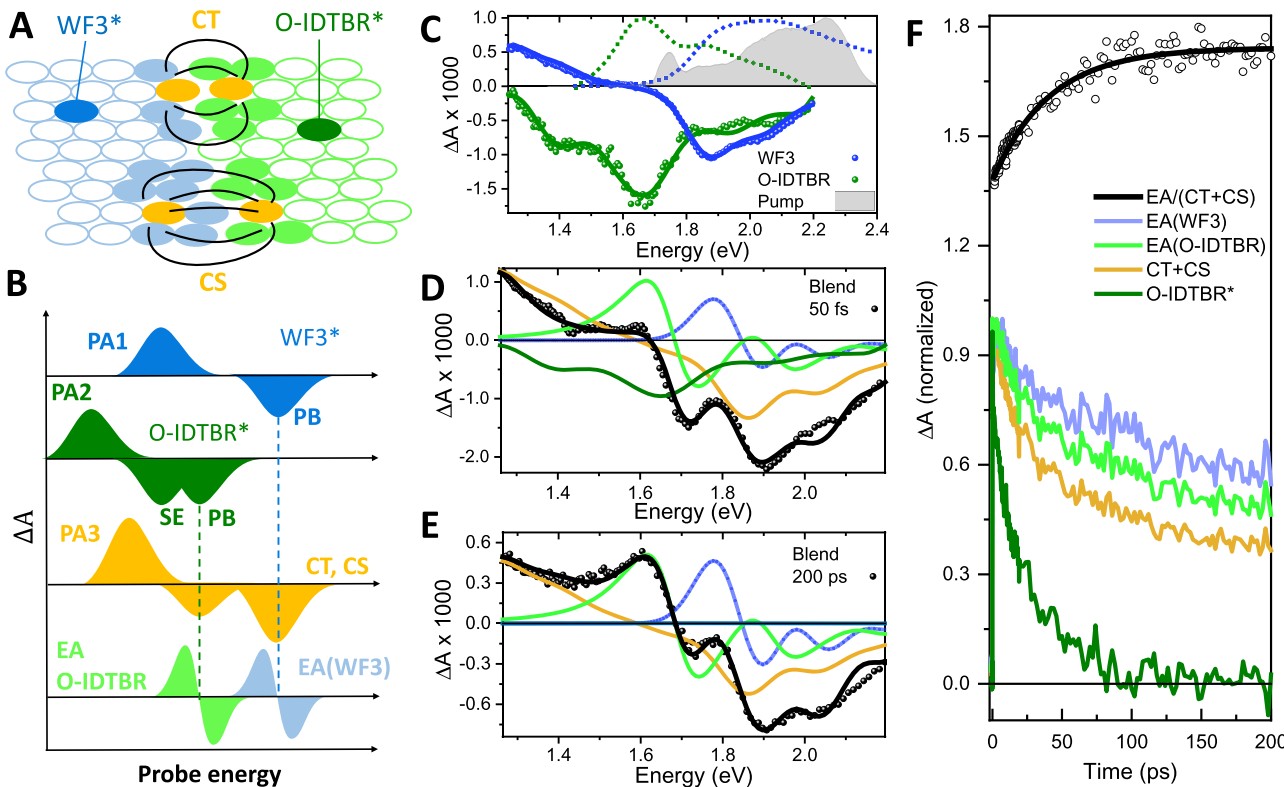

**Fig. 2 Ultrafast spectroscopy of the OPV blend. A** Schematic representation of states contributing to the TA spectra. Open shapes represent organic semiconductors in the ground state. Coloured shapes are organic semiconductors either in the electronically excited state (green: emissive O-IDTBR exciton; blue: WF3 polaron pair), or in a state created by a charge transfer event (gold: interfacial charge transfer state or charge-separated state), or in the electronic ground state but exposed to field lines of a charge pair, exhibiting a Stark shift of the fundamental electronic transition (light blue for WF3, light green for O-IDTBR). **B** Contributions to the TA spectra of these states. Identical bands occurring in several states are connected by dashed vertical lines. **C** TA spectra (dots) and fit functions (solid lines): the spectra were recorded 20 ps after excitation with 10 fs broadband pulses at parallel polarization with the probe pulses for films of pristine WF3 (blue) and annealed O-IDTBR (green). Dotted lines correspond to the normalised linear absorption and the shaded grey area shows the excitation spectrum. **D** TA of the WF3:O-IDTBR blend at 50 fs delays following photoexcitation (black dots), along with the fit (black line) and its components, following the same colour code as in panel (**B**). **E** Same as (**D**) for 200 ps after photoexcitation. **F** Dynamics of the concentration of each species included in the spectral modelling of the WF3:O-IDTBR blends (see Supplementary Information). Black dots correspond to the ratio EA/(CT + CS) that enables tracking the first jump of a charge away from the interface. The black solid line is an exponential fit with a 40.5 ps rise time.

available for CT generation, allowing the formation of non-equilibrium, band-like states[31]. The observation is in agreement with a recent demonstration of ultrafast electron transfer across the donor–acceptor interface after exciting donors in NFA-blends[32]. Since the formation of CT states from donor excitons outperforms all possible deactivation channels, it follows that the photoexcitation dynamics of our systems only depend on the HOMO–HOMO offsets; the LUMO–LUMO offsets do not contribute quantitatively as long as it is above a certain threshold value allowing the observed ultrafast transfer[33].

The spectral decomposition, as shown in Fig. 2D, yields a small initial population of acceptor excitons (dark green spectrum). The relative strength of this contribution approximately agrees with the expected amount of resonantly created acceptor excitons (about 30%), obtained by multiplication of the pump pulse spectral density with the pure donor and acceptor absorption spectra (Supplementary Information, Part 10). Therefore, we can associate the initial population of O-IDTBR excitons with resonant creation, showing that ultrafast energy transfer from donor to acceptor excitons does not play a significant role in our samples.

It has been shown that the ratio EA/(CT + CS) between the strength of the EA signal and the time-dependent densities of CT and CS states can be used to trace the separation of CT states into

CS states[34]. To find EA/(CT + CS), we decompose the TA spectra of the WF3:O-IDTBR blend in Fig. 2D, E into contributions from O-IDTBR excitons (green), CT/CS states (gold), and EA contributions from the WF3 (light blue) and O-IDTBR (light green), using all spectral shapes of these states available from ground-state absorption, photoluminescence (PL) and TA spectroscopy (for details see Supplementary Section 2). Repeating this analysis for all pump-probe delays, we obtain the time-dependent spectral weight of these contributions as shown in Fig. 2D, E with the same colour scheme from Fig. 2A, B. Figure 2F shows the normalised kinetics of the different states, and the EA/(CT + CS) ratio (shown in black) presents an exponential build-up with a time constant of 40.5 ps. A simple electrostatic simulation of the electric field caused by two charges across a planar D–A interface (see Supplementary Section 5) shows that this monoexponential rise can be assigned to the first jump away from the interface because further increases in the distance between the charges have a negligible effect on the EA signal (see Supplementary Fig. 20). Note that the first jump already creates charge-separated states for which the back-electron transfer to the ground state is strongly suppressed.

It is worth mentioning that charge separation kinetics in BHJ were previously studied employing pump-push photocurrent techniques[25,35]. However, it was recently shown that the push

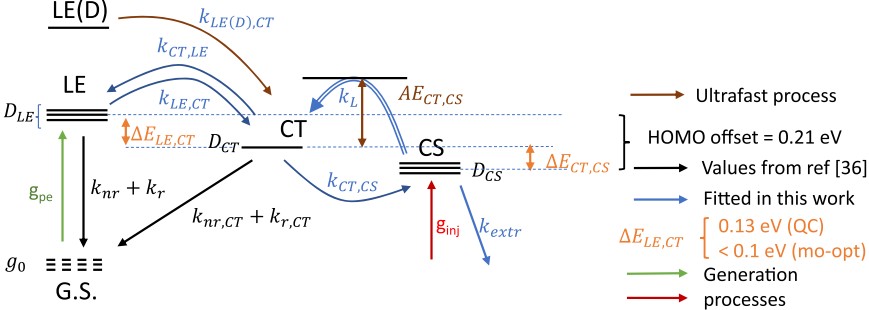

**Fig. 3 Photophysical model deployed for multi-objective optimization.** Black arrows: known values from the literature; blue: values optimized in this work; orange: driving forces for exciton splitting ($\Delta E_{LE,CT}$) and charge separation ($\Delta E_{CT,CS}$), estimated from quantum chemistry (QC) and multi-objective modelling. Bimolecular processes given as double lines. A number of horizontal lines indicate relative state abundance $g$ (not to scale). Dashed horizontal lines refer to states not explicitly included in the rate equation scheme. The states are generated either by optical excitation (arrow labelled $g_{pe}$) or by charge injection (arrow labelled $g_{inj}$), and $AE_{CT,CS}$ is the formal activation energy of CS formation in the Marcus pictures.

pulse does not cause enhanced CT dissociation in optimized blends where CT separation is only weakly activated[36]. The measured 40.5 ps charge separation time suffices to achieve 90% of charge separation yield, given the exciton lifetime of 530 ps of O-IDTBR LE states. This explains the high EQE of over 80% found in WF3:O-IDTBR, a system with very low HOMO–HOMO offset.

An interesting point is the decay of acceptor excitons on the same time scale as the build-up of the CS state (compare green and black curves in Fig. 2F). This is a direct consequence of the LE ⇔ CT equilibrium: the femtosecond pump pulse predominantly generates excitons in the donor phase. Due to the high driving force for exciton splitting, they convert instantaneously into CT states. However, acceptor LE states are accessible from CT states, and therefore an equilibration must occur. If the equilibration time is much shorter than the charge separation time, then both equilibrated species (LE and CT) are depleted on the same time scale by the charge separation process. An alternative explanation, namely diffusion of residual acceptor excitons followed by dissociation, can be excluded because it would not lead to monoexponential decay kinetics.

We note that triplet states have not been included in our model because the quantitative agreement between fit and experiment was achieved without considering them. A recent study has shown that triplet states do form upon CS recombination, however, they are relatively stable against deactivation (the main channel being triplet-triplet annihilation, which occurs predominantly at high excitation fluences)[37]. We have performed a comparative study using narrowband excitation exclusively in resonance with the acceptor or the donor exciton (Supplementary material, Part 6). These studies are qualitatively in agreement with the results of Fig. 2 albeit with a much longer instrumental response function, due to the narrowband pump pulses. We also observed a possible diffusion limitation of hole transfer after resonantly exciting the acceptor exciton; this has been found recently and ascribed to a lower wavefunction extension in the small molecule NFA system as compared to the donor polymer[32]. For this reason, we did not use the data from the narrowband experiment for kinetic modelling.

We complement the above experiments with time-dependent density functional theory (TDDFT) calculations to gain insight into the electronic structure of the WF3:O-IDTBR interface, modelled as a dimer formed by one repeat unit of WF3 and one O-IDTBR molecule (for details see Supplementary Section 4). Supplementary Fig. 15 shows the frontier molecular orbital wavefunctions for isolated WF3 and O-IDTBR monomers and for the blend of the two components (panels a,b, and c, respectively). Since the HOMO energies for the isolated WF3 and O-IDTBR are

very similar, we find a high degree of hybridization of these orbitals in the HOMO and HOMO-1 of WF3:O-IDTBR. This hybridization transfers significant oscillator strength from the LE state to the CT state so that the CT state acquires 1/5 of the oscillator strength of the LE, a relatively high value. In consequence, the energy of the CT state is also influenced; the calculation yields an energy offset $\Delta E_{LE,CT}$ of about 0.13 eV. Given the overall driving force (HOMO offset) of 0.21 eV, this value suggests a positive driving force for charge separation. However, there is substantial uncertainty in this value due to the limitations of the calculation, describing the interaction between a single donor monomer and a single acceptor molecule.

**Multi-objective modelling.** In order to understand the combination of high EQE and low $V_{OC}$ losses in WF3:O-IDTBR, we performed a multi-objective optimization, aiming to reproduce as closely as possible the three main results of this work (EQE > 80%, EL of blend indistinguishable from that of the pure acceptor, charge separation time 40 ps) by the simplest possible numerical model, taking into account both equilibria LE < = > CT and CT < = > CS and including radiative and non-radiative loss channels from LE and CT states. The model is given in Supplementary Fig. 12.

In our calculation, we consider five free parameters: the exciton breaking rate $k_{LE,CT}$, the driving force for exciton dissociation, $\Delta E_{LE,CT}$, the activation energy for charge separation $AE_{CT,CS}$, the bimolecular Langevin recombination constant $k_L$, and the charge extraction constant $k_{extr}$ (Fig. 3). The equilibrium CS ⇔ CT is given by the energy difference of these states, which is subject to the constraint $\Delta E_{CT,CS} + \Delta E_{LE,CT} = \Delta E_{HOMO}$; therefore the position of the CS ⇔ CT equilibrium cannot be varied independently from that of the LE ⇔ CT equilibrium; but the equilibration time is an additional degree of freedom, controlled by the formal activation energy of CS formation in the Marcus picture, $AE_{CT,CS}$, and $k_L$. Radiative and non-radiative decay rates of both LE (O-IDTBR) and CT (WF3:O-IDTBR) states are known from a recently published work[38]. The other parameters are coupled via the Boltzmann equilibrium. In the model, we found that $k_{LE,CT}$ plays no role as long as it is higher than $10^{10}$ s$^{-1}$, a condition safely met in highly efficient D–A blends;[32,38] therefore, we fixed this value to $10^{11}$ s$^{-1}$, so that the remaining number of free parameters is four. We highlight that the charge separation time of 40 ps has been obtained in a thin film without extraction layers, while EQE was measured at the short circuit and hence under an extraction field. As the charge separation time refers only to the first step of charge separation, we do not expect a significant influence of an extraction field of the charge separation time.

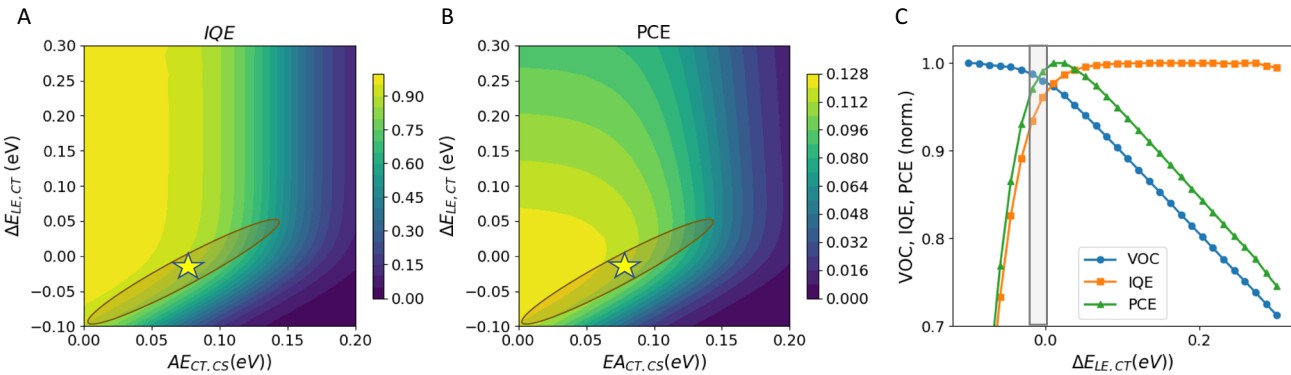

**Fig. 4 Multi-objective modelling. a** IQE and (**b**) PCE as a function of the driving force for exciton splitting $\Delta E_{LE,CT}$ and the activation energy for charge separation $AE_{CT,CS}$ shown in a false colour scale in a 30 × 30 grid. At each grid point, the charge extraction and Langevin constants were optimized so that the numerical model (see Supplementary Information, Eq. (S17)–(19)) best reproduced the three experimental observations: yellow star marks global optimum; orange stripe marks valley of local optima. **c** Vertical cut through panels (**a**) and (**b**) and of the corresponding graph for $V_{OC}$, shown in Supplementary Fig 13, at $AE_{CT,CS} = 0.075$ eV, normalised to the respective maxima. Grey rectangle marks the position of the local minimum for the multi-objective optimization, showing that the CT energy of WF3:O-IDTBR is close to the theoretical optimum position, given its specific LE and CT decay constants. This graph is similar for any position along the orange valley.

The result is shown in Fig. 4. The best match of the three objectives is found at a position marked by a yellow star, suggesting that the driving force for exciton splitting is close to zero. However, the orange trace in Fig. 4a, b indicates regions in which the match of the three objectives is nearly as good as in the global optimum. Therefore, we can only give an upper limit to $\Delta E_{LE,CT}$, which is at about 0.03 eV. Both methods, i.e., quantum chemistry and multi-objective modelling, thus agree that the CT state is higher in energy than the CS state, making charge separation an exothermic process in order to explain all experimental observations.

This result is counter-intuitive at first sight, given the stronger Coulomb attraction of a CT state as compared to a CS state and suggests that charge separation in low driving force systems should be endothermic because these systems lack excess energy needed for coherent long-range phenomena. Recently, temperature-dependent pump-push-probe transient data were presented in low driving force NFA blends, showing very slow charge separation kinetics in the hundreds of picoseconds time domain, supporting the picture of endothermic charge separation[39]. However, there is an increasing body of evidence that many systems display exothermic charge separation. For example, in similar IDTBR-based blends[36], it was suggested that an intermixed donor–acceptor interface can increase the CT energy thus providing a driving force towards the bulk leading to energy-dependent charge dynamics on the picosecond timescale[40]. Very recently, it has been shown that interfacial quadrupole moments can provide an additional driving force for CT dissociation[33,41]. Entropy contributions, on the other hand, can be controlled by optimizing the effective interfacial area[42]. Finally, by charge extraction experiments, charge separation without activation barrier has been demonstrated, another indication that charge separation is an energetically downhill process[41].

Although the multi-objective optimization does not yield a precise value for the driving force of exciton splitting, it does explain why WF3:O-IDTBR has high IQE and low voltage losses simultaneously. The area enclosed by the yellow ellipse in Fig. 4b, which is our best estimate of the combination of driving force and activation energies for WF3:O-IDTBR, is very close to the maximum achievable PCE for each given value of $AE_{CT,CS}$. Figure 4c shows vertical cuts through panels a and b and of the corresponding graph for $V_{OC}$ (Supplementary Fig 13) at $AE_{CT,CS} = 0.1$ eV, normalised to the

respective maxima. It is found that the $V_{OC}$ saturates for lower driving forces, while the IQE saturates for increasing driving forces. Hence, maximum PCE is expected at the crossing point between these normalised curves. The grey shaded region in Fig. 4c marks the most probable driving force for WF3:O-IDTBR, found by multi-objective optimization. It is found that the driving force, and thus the CT energy, of WF3:O-IDTBR, is very close to the theoretical optimum, with a PCE increase lower than 5% predicted upon further $AE_{CT,CS}$ optimization. Similar results are obtained for all values of the activation energy along the orange shaded region in Fig. 4a, b. Therefore, we can conclude that the reason for the high performance of WF3:O-IDTBR, combining high IQE and low $V_{OC}$ losses, is an optimized CT energy.

Our model is generally valid for donor–acceptor blends of the low driving force so that an equilibrium LE-CT is formed. As such, it has a significant impact on material search and device optimization in OPV. Consider the case of Y6, also boasting high IQE and low voltage losses in some of its blends[41]. This NFA is designed to be particularly rigid, thus combining good charge transport properties with long exciton lifetimes. It can be shown that long exciton lifetimes shift the normalised IQE (orange curve in Fig. 4c) to the left: the material can thus keep unity exciton splitting efficiencies for even lower driving forces[38]. This means that the maximum of the resulting normalised PCE (green curve in Fig. 4c) will be much broader for Y6 than for O-IDTBR in a given blend, so that for Y6, adjusting the CT energy to an optimum value is less critical for maximising PCE as it is for O-IDTBR. By predicting whether a dedicated morphology optimization will lead to a PCE gain in a given material combination, the application of our model can be used to avoid a tedious trial and error process. In this way, it broadens the range of possible candidate materials by lowering the PCE threshold for consideration.

Both $\Delta E_{LE,CT}$ and $\Delta E_{CT,CS}$ can be influenced by interfacial engineering which allows us to adjust them to obtain optimum IQE given the specific decay properties of the CT state in the blend[36]. Interfacial engineering can exploit both energetic and entropic effects.

## Discussion

In conclusion, we have demonstrated a new blend WF3:O-IDTBR showing very low non-radiative losses and at the same time an EQE of more than 80% in the spectral region of the low driving force. Using femtosecond TA spectroscopy, we have shown that

the CT state dissociates with a 40.5 ps time constant, which is slower than in previously measured systems with a high driving force, but sufficient to explain an EQE of 80%, despite the low CT decay rate. Numerical simulations using input from quantum chemistry successfully reproduce all experimental observations. We have shown that the reason for the high EQE, despite a low overall driving force, lies in a near-optimal distribution of this energy between the exciton dissociation and charge separation steps. From general considerations of the early CT processes, we propose that for each blend with given LE and CT lifetimes, there exists an optimal value for the driving forces for exciton dissociation and charge separation, that can be adjusted independently from the HOMO level alignment by exploiting interfacial energy destabilization and CT hybridization.

## Methods

**Materials**. WF3 and O-IDTBR were synthesized as reported elsewhere[8].

Fabrication of photovoltaic devices: pre-structured indium tin oxide (ITO) substrates were cleaned with acetone and isopropyl alcohol in an ultrasonic bath for 10 min each. After drying, the substrates were spin-coated with 40 nm of zinc oxide (ZnO) and different active layer based on 20 g L$^{-1}$ solution in chlorobenzene (CB). To complete the fabrication of the devices 10 nm of MoOx and 100 nm of Ag were thermally evaporated through a mask (with a 10.4 mm$^2$ active area opening) under a vacuum of ~$1 \times 10^{-6}$ mbar.

Current density–Voltage (J–V) measurements: the J–V characteristics were measured using a source measurement unit from BoTest. Illumination was provided by a solar simulator (Oriel Sol 1 A, from Newport) with AM1.5G spectrum at 100 mW/cm$^2$. UV–Vis absorption was performed on a Lambda 950, from Perkin Elmer. EQEs were measured using an integrated system from Enlitech, Taiwan. All the devices were tested in ambient air.

Sensitive EQE measurements (EQE$_{PV}$): EQE$_{PV}$ was measured using a modified Vertex 70 FTIR spectrometer from Brucker optics, equipped with a QTH lamp, quartz beam splitter, and external detector. A low noise current amplifier (Femto, DLPCA-200) is used to amplify the photocurrent produced upon illumination of the photovoltaic device with light modulated by the FTIR. The output voltage of the current amplifier is fed back to the external detector port of the FTIR, in order to be able to use the FTIR's software to collect the photocurrent spectrum.

EL: EL measurements were performed by using a chopper and applying a constant current (100 mA/cm$^2$) supplied by an external current/voltage source through the devices which have an active area of 0.104 cm$^2$. The emitted light then collected by a monochromator and detected by a liquid-nitrogen-cooled InGaAs detector. The spectrum was recorded by a standard lock-in technique. The system was wavelength calibrated.

TA: the TA setup is fed by a 100-fs, 1-kHz repetition rate Ti:sapphire laser system (Libra, Coherent) with a central wavelength of 800 nm. The ≈10 fs pump pulses span the spectral region from 1.7 to 2.4 eV and are generated by a non-collinear optical parametric amplifier (OPA), compressed to the near-transform limited duration at the sample position with a pair of chirped mirrors[43]. The probe pulses consist of a white-light continuum (WLC) generated by tightly focusing the output of an OPA at 1225 nm on a 2 mm thick sapphire plate, generating a WLC extending from 1.3 to 2.2 eV, limited on the NIR side by the short pass filter used to reject the residual 1225 nm OPA light[44]. Measurements are performed in transmission using an SP2150 Acton, Princeton Instruments spectrometer equipped with a CCD detector. The differential transmission ($\Delta T/T$) spectrum is acquired as a function of probe photon energy and pump-probe delay, and pump and probe beams are collinearly polarized.

**Reporting summary**. Further information on research design is available in the Nature Research Reporting Summary linked to this article.

## Data availability

The data that support the findings of this study are available on request from the corresponding author (N.G.).

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

## Acknowledgements

N.G. acknowledges the Imperial College Research Fellowship scheme. C.J.B. gratefully acknowledge funding from the Deutsche Forschungsgemeinschaft (DFG, German Research Foundation), Project no. 182849149-SFB 953. C.J.B. gratefully acknowledges financial support through the "Aufbruch Bayern" initiative of the state of Bavaria (EnCN and SFF) and the Bavarian Initiative "Solar Technologies go Hybrid" (SolTech) and funding from DFG project DFG INST 90/917. C.C.L. thanks the European Union for the financial support. G.C. acknowledges the support from the PRIN 2017 Project 201795SBA3—HARVEST.

## Author contributions

N.G. and C.J.B. conceived and developed the ideas. N.G. designed the experiments and performed device fabrication, electrical characterisation and data analysis. N.G. and M.S. performed EL, PL and FTPS measurements. F.V.A.C. performed TA measurements under the supervision of T.N., L.L. and G.C. S.F. performed the molecular quantum-chemical calculation under the supervision of A.G. L.L. developed the modelling of the TA, EQE, EL spectra. A.C. performed transient PL measurements. C.C. synthesized WF3. S.R. performed EL measurements under the supervision of D.N. A.W. synthesized the IDTBR acceptor under the supervision of I.M. N.G., F.V.A.C., and L.L. wrote the manuscript. M.S., D.N., C.J.B., D.B., V.G. and D.N. contributed to the revision of the manuscript. The projects were supervised by G.C., C.J.B. and L.L.

## Competing interests

The authors declare no competing interests.
