## [Peer Review File · Nature Communications]

Reviewers' comments:

Reviewer #1 (Remarks to the Author):

The authors demonstrated an organic photovoltaic (OPV) blend with both small voltage losses and high internal quantum efficiency (80%). They then combined spectroscopic investigations and numerical simulations (with inputs from quantum chemistry), trying to explain the high performance of this OPV blend. Their main finding (and also the key novelty of this manuscript) is that this unique blend has an optimal adjustment of the charge-transfer state energy. As a result, this blend shows optimal distribution of the HOMO offset for exciton dissociation and charge separation.

Although I believe that this work is potentially interesting, I think that it is too preliminary to be considered for Nature Communications. Specifically:

1, The demonstration of small voltage losses and efficient charge generation is not new. For example, there are quite several blends based on the Y-series acceptors showing similar features.

2, There are also several recent publications explaining the mechanisms behind small voltage losses and efficient charge generation in blends based on Y-series acceptors. Here are a few examples: 10.1038/s41566-019-0573-5 (this work emphasizes the role of low energetic disorder); 10.1002/adma.201906763 (this work emphasizes the role of the electrostatic interfacial field); 10.1002/adma.201903868 (this work emphasizes the role of optimal morphology). I consider that this manuscript provides yet another explanation (on a different blend). However, in order to convince the community that the optimal energy distribution is critical, the authors need to carefully design the experiments. For example, they can compare a few blends, by fixing one component (e.g., the acceptor) and gradually shifting the energetic levels of the other component. By performing this comparison, they might be able to demonstrate the critical role of the energy distribution. And they might also be able to see some trend between different blends. After that, they can go for simulations to gain deep insights.

Without proper design of the experiments, I do not think that this manuscript provides convincing understanding of a new mechanism.

Reviewer #2 (Remarks to the Author):

In this manuscript, the donor:acceptor blend WF3:O-IDTBR for organic photovoltaics is studied. This is a relevant model system for the currently high performance material combinations. Device and spectroscopic studies are used to model the thermodynamics of charge separation and recombination. It is shown that the relatively high performance of WF3:O-IDTBR arises from an optimal adjustment of the CT state energy, balancing competition between exciton and CT state decay and charge separation. A kinetic model shows how even at low offsets between CT state energies and exciton energies, high internal quantum efficiencies can be reached, which are thus combined with high voltages because of a high CT state energy.

In the abstract, authors further mention that they will discuss how interface engineering and donor-acceptor hybridization can be used to optimize device performance, but this part remains rather vague in the main text with no concrete new design rules formulated. Some already known design rules are mentioned. The introduction initially focusses on low non-radiative recombination losses and their need to achieve high efficiencies. Authors write: "Therefore, a complete physical understanding of the interplay between state hybridization, non-radiative voltage losses and fill factors is essential for the development of next generation OPV technologies." Further down in the text authors also write "it is therefore important to understand the reason for the observed combination of low non-radiative losses and high fill factor in the WF3:O-IDTBR blends."

However, the focus of the paper does not seem to be on these issues. The focus of the paper is on answering the important question why high CT state energies, close to the exciton energies, can still result in a high yield of free carrier generation. The fill-factor is not really mentioned anymore in the remainder of the text after the introduction and no new specific design rules on how to reduce non-radiative losses are deduced in this work. Hybridization between excitons and CT states to reduce non-radiative losses has been published earlier.

Because of the nice spectroscopic data and the new proposed kinetic model explaining why low driving force systems can still achieve high internal quantum efficiencies, I recommend publication of a revised manuscript, but, because of the reasons mentioned above, feel that some statements made in the introduction is a bit detached from the main findings.

Further comments:

- A key assumption or starting point for the analysis of the paper is the statement that: "It is well-established that HOMO-LUMO gaps are increased near the interface due to the broken symmetry, so the LE energies of both donor (blue) and acceptor (green) rise." However, no reference of this statement is given. One would expect references to several references if this statement is indeed well established.

- n-type semiconductors are mentioned in the abstract. These electron acceptors are not deliberately doped, so its inappropriate to call them n-type.

- Local triplet states are not detected or found to play a role. This might be worth mentioning, since the triplets on the acceptor are expected to be the lowest energy states.

Reviewer #3 (Remarks to the Author):

The authors present a study of the photophysical processes taking place in a polymer:NFA blend, in order to explain why it can simultaneously have high EQE and low non-radiative voltage losses. They relate this to the energy alignment of the acceptor LE state, the interfacial CT state and the CS state and an equilibration of the populations between those states. Although the abstract sounds interesting and the experiments are very carefully performed (I complement the analysis of the TA data), the manuscript does not live up to its promises and does not meet the standard for Nature Communications. The main conclusions are not sufficiently substantiated, because they are based on a kinetic model that entirely relies on unjustified assumptions. Moreover, concepts are confused (e.g. equilibrium/hybridization) and the paper is hard to read, requiring to consider the 112-page SI in order to understand what the authors did. The abstract gives the impression that work about interfacial engineering will be presented, but there is only a brief literature-based discussion and in the end only one system is presented, lacking the generality necessary for broad impact. I therefore suggest clarifying the manuscript (also by including details from the SI) and submitting to a more specialized journal.

Specific points:

1) "... how the HOMO level offset is distributed between driving exciton dissociation and charge separation" is a confusing statement in the abstract.

2) The introduction talks about the optimization of the FF, but the manuscript never comes back to this and rather talks about EQE, which is a different concept.

3) The EQE spectra in 1d have a suspicious shape that does not relate at all to the absorption. The authors should do TMM simulations and estimate the absorption within the device (and maybe IQE).

4) Unlike claimed in the manuscript, there is nothing in the SI to show how EQE_EL was obtained.

5) The exact same shape of the EL in the NFA and the blend shows that all EL stems from the acceptor. This does not necessarily imply equilibration between the two states nor that they have the same energy. Could charges get stuck in neat O-IDTBR domains and undergo recombination there, rather than at the interface?

6) Why was TA carried out with parallel polarization, this can introduce anisotropic effects that modify the dynamics?

7) Is the assignment of the increasing EA signal to CT->CS unambiguous? Could it be due to delayed charge formation after exciton diffusion (given the slow decay of the excited NFA)? Also, changes of the absorption spectrum in different regions of the blend have been shown to lead to time-resolved changes in the EA. This is the most important finding of the paper, the authors need to discuss alternative interpretations.

8) Very misleadingly, Figure 3 is completely speculative and based on previous literature (or not even), rather than derived from the results of the authors. In particular, the interface destabilization of the excited states is far-fetched (the authors claim that it is well established but do not even cite a reference). Due to bulk and interface molecular packing, microelectrostatics etc., the exciton energy at the interface can be unchanged, stabilized or destabilized depending on the system.

9) The hybridization is a very significant result but is only mentioned in one sentence in the main text. The outcome of the DFT calculations should be shown and discussed in more detail, including how the hybridization occurs in terms of energy, orbital overlap, electronic coupling etc. Also, how does this hybridization relate to the LE-CT equilibrium discussed in the rest of the manuscript? These are two different concepts. Hybridization means one mixed state, while an equilibrium is a conversion between two distinct states. The kinetics are different, and the two situations should not be confused.

10) The assumptions of the kinetic model must be discussed in the main text. Even in the SI, it is difficult to follow the details of the model and how the rate constants were obtained (how is the equilibration time found experimentally, what is $k_{t,CT}$, where is Scheme 1?). It seems that there is a high number of parameters and an even higher number of assumptions, while in the end, only part of the data is fitted (estimated IQE, EL spectrum, CS rise), but not the entire dynamics of the different species from TA. Therefore, it is not surprising that the many parameters of the model can fit a few data points. I'm certain that a simpler model would have allowed the same, so the authors need to disprove other scenarios.

11) In particular, the authors assume that there is an equilibrium between LE-CT and CT-CS, which depends according to eq. S25 on the free energy difference between the states. All the interpretation is based on this assumption, but there is absolutely no experimental evidence that such an equilibrium really exists. For example in the TA data, the slow decay of the acceptor exciton could simply be due to slow "one-way" splitting of the exciton and there is no sign of an offset due to an equilibrium LE population. The equilibrium between CT and CS also cannot be evidenced, as they have the same spectral signature. The authors need to show stronger evidence that the back reactions leading to the equilibria really take place. In the complex energetic landscape of a bulk heterojunction, this is not obvious.

12) The authors state that the equilibria take place between states in the bulk (LE, CS) and at the interface (CT), meaning that there is diffusion/transport of excitons/charges between the states. How is this taken into account in the model? How will it affect equilibration?

13) Some of the parameters are taken in un-contacted films (TA, I presume), while others come from device measurements. Whether there is extraction or not will however strongly affect the equilibria that the authors talk about. How will the presence/absence of extraction affect the CS rise that is fixed here from TA?

14) The reason of low EQE with a small LE-CT offset is that (according to e.g. Marcus theory), the exciton dissociation becomes slow (although recent work shows this is not necessarily the case). This is not discussed at all in the manuscript, and the energetics is assumed to only affect the equilibrium between the two states. However, the slow acceptor exciton decay (compared to instantaneous excited donor splitting) points to different exciton splitting times. In general, what do the authors expect to change when the acceptor or donor is initially excited (they mention excess energy in the CT state but do not pursue this further)?

15) Data with PCBM is initially shown, but this is then not exploited further in the data interpretation. Using the kinetic model (or an alternative one) on this system (with largely different energetics and a clearly distinct CT state) would make the story much stronger.

RESPONSE TO REFEREES

Reviewer #1 (Remarks to the Author):

The authors demonstrated an organic photovoltaic (OPV) blend with both small voltage losses and high internal quantum efficiency (80%). They then combined spectroscopic investigations and numerical simulations (with inputs from quantum chemistry), trying to explain the high performance of this OPV blend. Their main finding (and also the key novelty of this manuscript) is that this unique blend has an optimal adjustment of the charge-transfer state energy. As a result, this blend shows optimal distribution of the HOMO offset for exciton dissociation and charge separation.

Although I believe that this work is potentially interesting, I think that it is too preliminary to be considered for Nature Communications. Specifically:

Author response.

We thank the reviewer for the assessment that this work can be of interest. However, we do not share the reviewer's opinion that this work is preliminary. In the following we address why we believe that our claims are solid, rather than preliminary, and why we think that the insight gained is useful for the community.

Reviewer 1, point 1:

The demonstration of small voltage losses and efficient charge generation is not new. For example, there are quite several blends based on the Y-series acceptors showing similar features.

Author response.

We agree with the referee that there are other blends in organic photovoltaics also showing small voltage losses and efficient charge generation at the same time. However, it is not the goal of our paper to present the first ever reported blend showing these properties. Our first key claim is the experimental demonstration that CT state separation into free carriers occurs within 40 ps, which to our knowledge has not been measured for any other blend with low overall driving force. Our second key claim is that all our experimental observations (spectra and photoexcitation dynamics) can be reproduced by assuming an optimal CT energy which is below the exciton energy but **above** the energy of separate carriers. Our method is based on plain thermodynamic considerations. It is therefore generally valid and can be adapted to other donor:acceptor organic bulk heterojunction solar cells, by adjusting parameters such as exciton lifetime and reorganization energy. We therefore believe that our results are solid, general, and therefore useful.

In the revised version, we include a graph and a dedicated section better highlighting the general usefulness of our method: we show that PCE of Y6 does not depend as sharply on the CT energy as it does for O-IDTBR. We conclude that our model broadens the range of accessible material combinations (by predicting whether or not morphology optimization is promising) thereby avoiding trial and error and allowing a more targeted approach in OPV technology.

changes: New text on pg. 4

“We show that this equilibrium is decisively controlled by the degeneracy and energy of the interfacial CT states, and that it attains a near-optimum value in WF3:O-IDTBR, explaining its ability to produce high open circuit voltage(V_{oc}) and EQE at the same time. We generalize these results by showing that, for any blend with low driving force, there exists an optimum CT energy, which is at the crossing point of normalized V_{oc} and EQE curves along the CT energy axis. We discuss known methods to adjust the CT energy in a given blend in light of this new functional relationship.”

changes: New text on pg. 15

“Our model is generally valid for donor-acceptor blends of low driving force, so that an equilibrium LE-CT is formed. As such, it has significant impact for material search and device optimization in OPV. Consider the case of Y6, also boasting high IQE and low voltage losses in some of its blends.³⁸ This NFA is designed to be particularly rigid, thus combining good charge transport properties with long exciton lifetimes. It can be shown that long exciton lifetimes shift the normalized IQE (orange curve in Fig. 4c) to the left: the material can thus keep unity exciton splitting efficiencies for even lower driving forces.³⁶ This means that the maximum of the resulting normalized PCE (green curve in Fig. 4c) will be much broader for Y6 than for O-IDTBR in a given blend, so that for Y6, adjusting the CT energy to an optimum value is less critical for maximizing PCE as it is for O-IDTBR. By predicting whether a dedicated morphology optimization will lead to a PCE gain in a given material combination, application of our model can be used to avoid a tedious trial and error process. In this way, it broadens the range of possible candidate materials by lowering the PCE threshold for consideration.”

changes: Revised Figure 4.

Reviewer 1, point 2:

There are also several recent publications explaining the mechanisms behind small voltage losses and efficient charge generation in blends based on Y-series acceptors. Here are a few examples:

10.1038/s41566-019-0573-5 (this work emphasizes the role of low energetic disorder);

10.1002/adma.201906763 (this work emphasizes the role of the electrostatic interfacial field);

10.1002/adma.201903868 (this work emphasizes the role of optimal morphology). I consider that this manuscript provides yet another explanation (on a different blend).

Author response:

We thank the reviewer for this point. Indeed, there is agreement among all referees that the implications of our findings should be better explained. We share this view, and the references cited by the referee present an excellent starting point to do so. It turns out that our work can contribute to harmonize seemingly contradicting results in the literature. Karki et al, Adv. Mater. 2019, 31, 1903868, find that an abrupt donor-acceptor interface is beneficial for charge separation, while other authors such as Cha et al., Adv. Energy Mater. 2019, 1901254, show that the presence of both mixed and pure phases provides additional driving force for CT state separation.

Our model, rather than being “yet another explanation” as the referee claims, is in fact able to justify the findings from both authors: Y6 is designed to have a very low reorganisation energy and boasts a long exciton lifetime. The Boltzmann equilibrium between the exciton and CT states therefore will enable the barrierless charge carrier formation, observed by Perdigón-Toro et al., Adv. Mater. 2020, 32, 1906763.

In contrast, blends which involve the IDTBR series, such as our work but also that of Cha et al., do require a carefully adjusted CT energy which is above that of the charge separated states. Recall that one of the main selling points of organic photovoltaics will be their versatility. The community should therefore make sure that also other NFA than the Y series, should be driven to their performance limit. The approach adopted in our manuscript allows the prediction whether morphology optimization, which is a tedious process involving many experimental parameters, is promising for a given donor-acceptor combination.

Now the text reads as follows:

“Our model is generally valid for donor-acceptor blends of low driving force, so that an equilibrium LE-CT is formed. As such, it has significant impact for material search and device optimization in OPV. Consider the case of Y6, also boasting high IQE and low voltage losses in some of its blends.³⁸ This NFA is designed to be particularly rigid, thus combining good charge transport properties with long exciton lifetimes. It can be shown that long exciton lifetimes shift the normalized IQE (orange curve in Fig. 4c) to the left: the material can thus keep unity exciton splitting efficiencies for even lower driving forces.³⁶ This means that the maximum of the resulting normalized PCE (green curve in Fig. 4c) will be much broader for Y6 than for O-IDTBR in a given blend, so that for Y6, adjusting the CT energy to an optimum value is less critical for maximizing PCE as it is for O-IDTBR. By predicting whether a dedicated morphology optimization will lead to a PCE gain in a given material combination, application of our model can be used to avoid a tedious trial and error process. In this way, it broadens the range of possible candidate materials by lowering the PCE threshold for consideration.”

Reviewer 1, point 3:

However, in order to convince the community that the optimal energy distribution is critical, the authors need to carefully design the experiments. For example, they can compare a few blends, by fixing one component (e.g., the acceptor) and gradually shifting the energetic levels of the other component. By performing this comparison, they might be able to demonstrate the critical role of the energy distribution. And they might also be able to see some trend between different blends. After that, they can go for simulations to gain deep insights. Without proper design of the experiments, I do not think that this manuscript provides convincing understanding of a new mechanism.

Author response:

We thank the referee for this suggestion. Such studies have indeed been performed, see e.g. Qian et al., Nat. Mater. 17, 703–709 (2018), and they yielded important insight, for example on the role of exciton-CT hybridization. However, our goal is different: we want to find and control handles that optimize the

performance within one and the same material combination. Given this setting, we decided not to use a broad range of different material combinations, but rather maximize the knowledge gathered from a single material, in order to maximize the number of constraints a candidate model must fulfil to agree with all experimental observations.

We highlight that the same approach that we used, has been chosen by all three papers that referee 1 cites, namely gathering broad and complementary knowledge about one blend rather than scanning different blends.

Reviewer #2, point 1:

In this manuscript, the donor:acceptor blend WF3:O-IDTBR for organic photovoltaics is studied. This is a relevant model system for the currently high performance material combinations. Device and spectroscopic studies are used to model the thermodynamics of charge separation and recombination. It is shown that the relatively high performance of WF3:O-IDTBR arises from an optimal adjustment of the CT state energy, balancing competition between exciton and CT state decay and charge separation. A kinetic model shows how even at low offsets between CT state energies and exciton energies, high internal quantum efficiencies can be reached, which are thus combined with high voltages because of a high CT state energy.

In the abstract, authors further mention that they will discuss how interface engineering and donor-acceptor hybridization can be used to optimize device performance, but this part remains rather vague in the main text with no concrete new design rules formulated. Some already known design rules are mentioned. The introduction initially focusses on low non-radiative recombination losses and their need to achieve high efficiencies. Authors write: "Therefore, a complete physical understanding of the interplay between state hybridization, non-radiative voltage losses and fill factors is essential for the development of next generation OPV technologies." Further down in the text authors also write "it is therefore important to understand the reason for the observed combination of low non-radiative losses and high fill factor in the WF3:O-IDTBR blends."

However, the focus of the paper does not seem to be on these issues. The focus of the paper is on answering the important question why high CT state energies, close to the exciton energies, can still result in a high yield of free carrier generation. The fill-factor is not really mentioned anymore in the remainder of the text after the introduction and no new specific design rules on how to reduce non-radiative losses are deduced in this work. Hybridization between excitons and CT states to reduce non-radiative losses has been published earlier.

Author response:

We thank the reviewer for the generally positive assessment of the manuscript. Fill factor losses are explicitly treated in our numerical model by considering non-geminate recombination as well as geminate recombination, the latter causing predominantly J_{sc} losses. We agree with the referee that we have not discussed the results from this model in due detail. This has been improved in the revised version.

Now the text reads as follows:

“In our calculation we consider five free parameters: the exciton breaking rate $k_{LE,CT}$, the driving force for exciton dissociation, $\Delta E_{LE,CT}$, the activation energy for charge separation $EA_{CT,CS}$ the bimolecular Langevin recombination constant k_L , and the charge extraction constant k_{extr} . The equilibrium $CS \leftrightarrow CT$ is given by the energy difference of these states, which is subject to the constraint $\Delta E_{CT,CS} + \Delta E_{LE,CT} = \Delta E_{HOMO}$; therefore the position of the $CS \leftrightarrow CT$ equilibrium cannot be varied independently from that of the $LE \leftrightarrow CT$ equilibrium; but the equilibration time is an additional degree of freedom, controlled by $EA_{CT,CS}$ and k_L .”

Reviewer #2, point 2:

Because of the nice spectroscopic data and the new proposed kinetic model explaining why low driving force systems can still achieve high internal quantum efficiencies, I recommend publication of a revised manuscript, but, because of the reasons mentioned above, feel that some statements made in the introduction is a bit detached from the main findings.

Author response:

We thank the reviewer for this recommendation. In the revised version, we have streamlined the introduction to relate to our key claims more rigorously.

In the revised version the text reads as follows:

“However, a low CT stabilization energy $\Delta E_{LE,CT} = E_{LE} - E_{CT}$ can lead to incomplete exciton dissociation, thus penalizing the EQE(ref 24-25).”

Reviewer #2, point 3:

A key assumption or starting point for the analysis of the paper is the statement that: “It is well-established that HOMO-LUMO gaps are increased near the interface due to the broken symmetry, so the LE energies of both donor (blue) and acceptor (green) rise.” However, no reference of this statement is given. One would expect references to several references if this statement is indeed well established.

Author response:

The referee is right. In the revised version, we cite a work by Cha et al., Adv. Energy Mater. 2019, 1901254, in which it has been shown (in an NFA from the IDTBR series) that the increased exciton energy in interpenetrating donor-acceptor interfaces is caused by a higher torsional freedom than in the ordered bulk.

Now the text reads as follows:

“This result is counter-intuitive at first sight, given the stronger Coulomb attraction of a CT state as compared to a CS state; however, there is an increasing body of evidence that many systems display such behavior. For example, in similar IDTBR-based blends,³⁴ it was suggested that an intermixed donor-acceptor interface can increase the CT energy thus providing a driving force towards the bulk leading to

energy-dependent charge dynamics on the picosecond timescale.³⁷ Very recently, it has been shown that interfacial quadrupole moments can provide an additional driving force for CT dissociation. ref 38 Entropy contributions, on the other hand, can be controlled by optimizing the effective interfacial area. ref 39 Finally, by charge extraction experiments, charge separation without activation barrier has been demonstrated, another indication that charge separation is an energetically downhill process .ref 38”

Reviewer #2, point 4:

n-type semiconductors are mentioned in the abstract. These electron acceptors are not deliberately doped, so its inappropriate to call them n-type.

Author response:

The referee is right. In the revised version, we consistently use the term “electron acceptor” instead of n-type semiconductor.

changes: “The medium bandgap D-A copolymer WF3 was selected as electron donor material, while the electron acceptor was either the fullerene derivative PC70BM, or the small molecule NFA O-IDTBR (Fig. 1a).”

Reviewer #2, point 5:

Local triplet states are not detected or found to play a role. This might be worth mentioning, since the triplets on the acceptor are expected to be the lowest energy states.

Author response:

We thank the referee for highlighting this point. The question whether triplet states are being formed, and if yes, how they interconvert with other electronic states, is a very important one for both efficiency and stability considerations. Femtosecond transient absorption spectroscopy is not a technique for trace analysis, so we cannot exclude a presence of triplet states below the percent regime. In our spectral decomposition scheme, we reach quantitative fits without considering triplet states.

In the revised version, we include a recent paper by Karuthedath et al., showing that triplets can form in blends comprising NFA but they are relatively stable against interconversion, giving a positive perspective for organic photovoltaics at low energy offsets.

Now the text reads as follows:

“We note that triplet states have not been included into our model because quantitative agreement between fit and experiment was achieved without considering them. A recent study has shown that triplet states do form upon CS recombination, however they are relatively stable against deactivation (the main channel being triplet-triplet annihilation, which occurs predominantly at high excitation fluences). Ref 35 ”

Reviewer #3, point 1 (Remarks to the Author):

The authors present a study of the photophysical processes taking place in a polymer:NFA blend, in order to explain why it can simultaneously have high EQE and low non-radiative voltage losses. They relate this to the energy alignment of the acceptor LE state, the interfacial CT state and the CS state and an equilibration of the populations between those states. Although the abstract sounds interesting and the experiments are very carefully performed (I complement the analysis of the TA data), the manuscript does not live up to its promises and does not meet the standard for Nature Communications.

The main conclusions are not sufficiently substantiated, because they are based on a kinetic model that entirely relies on unjustified assumptions. Moreover, concepts are confused (e.g. equilibrium/hybridization) and the paper is hard to read, requiring to consider the 112-page SI in order to understand what the authors did.

Author response:

We thank the referee for generally supporting the scope of the paper and for the acknowledgment of adequate experimental design and high quality of the experimental data and their analysis. However, we cannot accept the notion that our model “entirely relies on unjustified assumptions”. This comment by referee 3 mainly refers to our notion of a thermodynamic equilibrium between states of similar energy. This is however not an unjustified assumption, but plain thermodynamics. As we note in the manuscript, it has been highlighted by other authors that such an equilibrium must be taken into account if the energy offset between LE and CT states becomes small (Qian et al., Nature Mat 2018). Moreover, some of us (Classen et al., Nature Energy 2020, DOI:10.1038/s41560-020-00684-7) have meanwhile demonstrated the presence of such a Boltzmann equilibrium in a broad range of donor-acceptor combinations, including WF3:o-IDTBR. Therefore, we insist that the foundations of our reasoning, namely the thermodynamic equilibrium between LE and CT states, are well supported by experimental evidence. In the revised version, we made an effort to justify all assumptions on which our kinetic model is based in the main text, rather than referring only to the supporting material.

changes: New Fig. 3; description of multi-objective optimization completely re-written on pg 12-15

Reviewer # 3, point 2:

The abstract gives the impression that work about interfacial engineering will be presented, but there is only a brief literature-based discussion and in the end only one system is presented, lacking the generality necessary for broad impact. I therefore suggest clarifying the manuscript (also by including details from the SI) and submitting to a more specialized journal.

Author response:

There is agreement among all referees that the discussion of the implications of our findings must be improved. We share this view. In the revised manuscript, we have provided a dedicated section to show

that our model is generally valid for D-A blends of low driving force, and we clearly exemplify the difference between WF3:O-IDTBR, and other high performance NFA such as Y6

We do not agree with the statement, made by reviewer 3, that presenting more than one system is a necessary condition for broad impact. In fact, all high-impact papers cited by referee 1, are only dealing with a single system.

Now the text reads as follows:

“Our model is generally valid for donor-acceptor blends of low driving force, so that an equilibrium LE-CT is formed. As such, it has significant impact for material search and device optimization in OPV. Consider the case of Y6, also boasting high IQE and low voltage losses in some of its blends.³⁸ This NFA is designed to be particularly rigid, thus combining good charge transport properties with long exciton lifetimes. It can be shown that long exciton lifetimes shift the normalized IQE (orange curve in Fig. 4c) to the left: the material can thus keep unity exciton splitting efficiencies for even lower driving forces.³⁶ This means that the maximum of the resulting normalized PCE (green curve in Fig. 4c) will be much broader for Y6 than for O-IDTBR in a given blend, so that for Y6, adjusting the CT energy to an optimum value is less critical for maximizing PCE as it is for O-IDTBR. By predicting whether a dedicated morphology optimization will lead to a PCE gain in a given material combination, application of our model can be used to avoid a tedious trial and error process. In this way, it broadens the range of possible candidate materials by lowering the PCE threshold for consideration.”

Reviewer #3, point 3:

1) “... how the HOMO level offset is distributed between driving exciton dissociation and charge separation” is a confusing statement in the abstract.

Author response:

We agree with the referee. In the revised version, we rephrased this statement to “... how the available overall driving force, specific for each material combination, is efficiently used to maximize both exciton splitting and charge separation”

In the revised version the text reads as follows:

“We combine device and spectroscopic data to model the thermodynamics of charge separation and extraction, revealing that the relatively high performance of WF3:O-IDTBR arises from an optimal adjustment of the CT state energy, which determines how the available overall driving force, specific for each material combination, is efficiently used to maximize both exciton splitting and charge separation.”

Reviewer #3, point 4:

2) The introduction talks about the optimization of the FF, but the manuscript never comes back to this and rather talks about EQE, which is a different concept.

Author response:

We agree with the referee. In the revised manuscript, we focus our reasoning on EQE losses (due to incomplete exciton splitting) and VOC losses (due to radiative and nonradiative recombination).

In the revised version the text reads as follows:

“However, a low CT stabilization energy $\Delta E_{LE,CT} = E_{LE} - E_{CT}$ can lead to incomplete exciton dissociation, thus penalizing the EQE(ref 24-25).”

Reviewer #3, point 5:

3) The EQE spectra in 1d have a suspicious shape that does not relate at all to the absorption. The authors should do TMM simulations and estimate the absorption within the device (and maybe IQE).

Author response:

We kindly disagree with the reviewers' comment. A similar EQE spectrum has been reported in ref 36. Moreover, the Supplementary Figure 16 of ref.36 depicts the IQE spectrum of WF3:O-IDTBR, which reaches values in the excess of 90%.

Reviewer #3, point 6:

4) Unlike claimed in the manuscript, there is nothing in the SI to show how EQE_EL was obtained.

Author response:

In the SI (pg. 49-50) we reported the experimental methodology for the calculation of EQE_EL as: EL measurements were performed by using a chopper and applying a constant current (100mA/cm²) supplied by an external current/voltage source through the devices which have an active area of 0.104 cm². The emitted light was collected by a monochromator and detected by liquid-nitrogen-cooled InGaAs detector. The spectrum was recorded by a standard lock-in technique. The system was wavelength calibrated. The calculations are based on the widely used method developed by Vandewal and co-workers (ref 2).

Reviewer # 3, point 7:

5) The exact same shape of the EL in the NFA and the blend shows that all EL stems from the acceptor. This does not necessarily imply equilibration between the two states nor that they have the same energy. Could charges get stuck in neat O-IDTBR domains and undergo recombination there, rather than at the interface?

Author response:

This is an interesting point. EL occurs after carrier injection through the contacts. In order to “get stuck” in neat o-IDTBR domains as the referee states, a hole must be transferred from the WF3 phase (into which it was originally injected) into the o-IDTBR phase. Given the small HOMO offsets of the two

components, such a transfer is indeed possible. We think however, that such a transfer cannot occur to a significant extent, because dwelling of both carrier types in the same phase would entail Langevin type recombination in homogeneous phase. In contrast, there is ample evidence that most organic photovoltaic blends show sub-Langevin recombination, because recombination is not a bulk process but occurs only at the donor-acceptor interface (M.C.Heiber et al., PRL114,136602 (2015)).

In the revised version, supplementary Figure 14, we display the Pareto Front of the multi-objective optimization. It is very close to the perfect match for all criteria, which suggests that the simple model, as chosen, is not overdefined but will be over-defined if another free parameter is incorporated. Therefore, we did not explicitly consider the inclusion of the above model.

Changes: revised version, supplementary Figure 14

Reviewer #3, point 8:

6) Why was TA carried out with parallel polarization, this can introduce anisotropic effects that modify the dynamics?

Author response:

We have performed femtosecond TA spectroscopy under both parallel and magic angle polarizations. As shown below, the data can be fitted with the same spectral model but the fits are significantly better for parallel polarization than for magic angle polarization. To accurately reproduce EA, the original spectral model describing ground state absorption must be used, and no adaptation of centre energies nor bandwidths is allowed. We were able to perform this exact procedure with parallel polarizations but not with magic angle polarizations. One possible explanation is that only under parallel polarizations, the spectral resolution of the TA data agrees exactly with that of ground state absorption.

The analysis of our data suggests that the increase of the electroabsorption feature points to the first step of charge separation. Therefore, anisotropy effects should be limited. We highlight that, in the presence of bimolecular kinetics, depolarisation effects cannot fully be avoided even working at magic angle polarization.

This discussion has been included into the revised version.

Figure 1. Femtosecond TA spectra (long-lived SVD component, equivalent to TA spectrum of pump-probe delay between 100 and 200 ps) of WF3:o-IDTBR after pumping with 10 fs broadband optical pulses, left: parallel polarization, right: magic angle polarization of pump and probe beams (black solid lines). Blue, orange and red lines give contributions from the CT/CS states, EA of acceptor, EA of donor, respectively, summing up to the black dashed line. Under parallel polarization, the spectral resolution seems slightly better so that the spectral shapes, predicted from ground state absorption, match the TA spectrum exactly.

Now the text reads as follows:

“Pump and probe beams have parallel polarization; we also performed the study at magic angle polarization, qualitatively showing similar results. For a quantitative evaluation, we preferred parallel polarization due to the higher signal quality”

Reviewer #3, point 9:

7) Is the assignment of the increasing EA signal to CT->CS unambiguous? Could it be due to delayed charge formation after exciton diffusion (given the slow decay of the excited NFA)? Also, changes of the absorption spectrum in different regions of the blend have been shown to lead to time-resolved changes in the EA. This is the most important finding of the paper, the authors need to discuss alternative interpretations.

Author response:

We agree with the referee that this finding is central to the argumentation of the paper and must therefore be duly justified. In fact, the decay of residual excitons in the acceptor phase and the buildup of CS states occurs roughly on the same time scale. But this is exactly what is expected if the equilibrium picture is true: The femtosecond pulse predominantly creates LE states in the donor, which due to high driving force convert instantaneously into CT states. As the CT states are in equilibrium with LE states in

the acceptor, a small population of LE states must form. If the equilibration kinetics $LE \leftrightarrow CT$ is much faster than charge separation, then LE and CT are depleted on the same time scale.

We exclude diffusional kinetics because we observe a monoexponential decay.

We consider this as a further confirmation of the equilibrium picture, and include this discussion into the revised manuscript.

Now the text reads as follows:

“An interesting point is the decay of acceptor excitons on the same time scale as the build-up of the CS state (compare green and black curves in Fig. 2f). This is a direct consequence of the $LE \leftrightarrow CT$ equilibrium: the femtosecond pump pulse predominantly generates excitons in the donor phase. Due to the high driving force for exciton splitting, they convert instantaneously into CT states. However, CT states are in equilibrium with acceptor LE states, therefore an equilibration with LE states must occur. If the equilibration time is much shorter than the charge separation time, then both equilibrated species (LE and CT) are depleted on the same time scale by the charge separation process. An alternative explanation, namely diffusion of residual acceptor excitons followed by dissociation, can be excluded because it would not lead to monoexponential decay kinetics.”

Reviewer # 3, point 10:

8) Very misleadingly, Figure 3 is completely speculative and based on previous literature (or not even), rather than derived from the results of the authors. In particular, the interface destabilization of the excited states is far-fetched (the authors claim that it is well established but do not even cite a reference). Due to bulk and interface molecular packing, microelectrostatics etc., the exciton energy at the interface can be unchanged, stabilized or destabilized depending on the system.

Author response:

We agree with the referee. Original goal for the inclusion of this graph was a pictorial summary of literature results bringing the CT energy above the CS energy, despite the Coulomb attraction. In the revised version, we have decided to replace this figure with a new figure (new figure 3) clearly bringing our findings into a general context of D-A blends with low driving force.

changes: new fig. 3

Reviewer #3, point 11:

9) The hybridization is a very significant result but is only mentioned in one sentence in the main text. The outcome of the DFT calculations should be shown and discussed in more detail, including how the hybridization occurs in terms of energy, orbital overlap, electronic coupling etc. Also, how does this hybridization relate to the LE-CT equilibrium discussed in the rest of the manuscript? These are two different concepts. Hybridization means one mixed state, while an equilibrium is a conversion between two distinct states. The kinetics are different, and the two situations should not be confused.

Author response:

We agree with the referee that the implication of the quantum chemical results on the rate equation model should be discussed more prominently. In the revised manuscript, we introduced a figure summarizing the most important findings of our quantum chemical modelling namely, a strong transfer of oscillation strength from the LE to the CT state, and a CT energy which is only about 0.13 eV below that of the LE state.

In the revised version, we also clearly distinguish a Boltzmann equilibrium between two different states with associated energies, from a single hybridized state.

Now the text reads as follows:

“We complement the above experiments with time-dependent density functional theory (TDDFT) calculations to gain insight into the electronic structure of the WF3:O-IDTBR interface, modelled as a dimer formed by one repeat unit of WF3 and one O-IDTBR molecule (for details see Supplementary Section 4). Supplementary Fig. 15 shows the frontier molecular orbital wavefunctions for isolated WF3 and O-IDTBR monomers and for the blend of the two components (panels a,b, and c, respectively). Since the HOMO energies for the isolated WF3 and O-IDTBR are very similar, we find a high degree of hybridization of these orbitals in the HOMO and HOMO-1 of WF3:O-IDTBR. This hybridization transfers significant oscillator strength from the LE state to the CT state so that the CT state acquires 1/5 of the oscillator strength of the LE, a relatively high value. In consequence, the energy of the CT state is also influenced; the calculation yields an energy offset $\Delta E_{LE,CT}$ of about 0.13 eV. Given the overall driving force (HOMO offset) of 0.21 eV, this value suggests a positive driving force for charge separation. However, there is substantial uncertainty in this value due to the limitations of the calculation, describing the interaction between a single donor monomer and a single acceptor molecule.”

Reviewer #3, point 12:

10) The assumptions of the kinetic model must be discussed in the main text. Even in the SI, it is difficult to follow the details of the model and how the rate constants were obtained (how is the equilibration time found experimentally, what is $k_{t,CT}$, where is Scheme 1?). It seems that there is a high number of parameters and an even higher number of assumptions, while in the end, only part of the data is fitted (estimated IQE, EL spectrum, CS rise), but not the entire dynamics of the different species from TA. Therefore, it is not surprising that the many parameters of the model can fit a few data points. I'm certain that a simpler model would have allowed the same, so the authors need to disprove other scenarios.

Author response:

We agree with the referee that the modelling procedure was hard to follow. In our opinion, what complicated things most, was that we performed the modelling in two steps. This inevitably led to confusion about which parameter is fixed or variable and under which circumstances.

In the revised version, we have completely redone the multi-objective optimization in a single step. This drastically reduces and simplifies the description in the supplementary material; it also makes absolutely clear which are the free and the fixed parameters, and which is the model deployed. in short:

- the model is the simplest possible model considering the equilibria between LE and CT, and between CT and CS, and deactivation of LE and CT states.

- deactivation rates of LE and CT states are known from a previous paper

- free parameters are: exciton dissociation rate, CT energy, activation energy of charge separation, Langevin recombination rate, charge extraction rate. That is, five free parameters.

- we have shown that the exciton dissociation rate plays no role as long as it is higher than 10^{10} s^{-1} , a condition safely met in low driving force systems. Therefore, the number of free parameters is exactly four.

- we avoid performing the multi-objective optimization as a “black box”, showing only the final result. In the revised version, we also show local optima besides the global optimum, and we show that our conclusions are valid even for the local optima.

- We finally demonstrate, by the occurrence of Pareto Fronts (New Supplementary Figure 14), that our model is not overdefined.

Finally, we note that feature extraction and dimensionality reduction are two fundamental principles that facilitate a multi-objective optimization; therefore, it is not useful to fit a dataset again for which we have already extracted a feature, namely a lifetime.

changes: completely re-written chapter in SI referring to multi-objective optimization;

Now the text reads as follows:

“In order to understand the combination of high EQE and low VOC losses in WF3:O-IDTBR, we performed a multi-objective optimization, aiming to reproduce as closely as possible the three main results of this work (EQE > 80%, EL of blend indistinguishable from that of the pure acceptor, charge separation time 40 ps) by the simplest possible numerical model, taking into account both equilibria $LE \rightleftharpoons CT$ and $CT \rightleftharpoons CS$ and including radiative and non-radiative loss channels from LE and CT states. The model is given in supplementary Figure 12.”

Reviewer #3, point 13:

11) In particular, the authors assume that there is an equilibrium between LE-CT and CT-CS, which depends according to eq. S25 on the free energy difference between the states. All the interpretation is based on this assumption, but there is absolutely no experimental evidence that such an equilibrium really exists. For example in the TA data, the slow decay of the acceptor exciton could simply be due to slow “one-way” splitting of the exciton and there is no sign of an offset due to an equilibrium LE population. The equilibrium between CT and CS also cannot be evidenced, as they have the same

spectral signature. The authors need to show stronger evidence that the back reactions leading to the equilibria really take place. In the complex energetic landscape of a bulk heterojunction, this is not obvious.

Author response:

We agree with the referee that solid justification for the equilibrium assumption should be presented. However, the notion of a thermodynamic equilibrium is not new; it has been highlighted by other authors that such an equilibrium must be taken into account if the energy offset between LE and CT states becomes small (Qian et al., Nature Mat 2018). Moreover, some of us have meanwhile unambiguously demonstrated such an equilibrium between LE and CT states for many blends including WF3:o-IDTBR (Classen et al., Nat. Energy 2020, DOI:10.1038/s41560-020-00684-7). An equilibrium between CT and CS states, albeit for different blends, has been demonstrated by the McGehee group (Burke et al., Adv. En. Mat 5(11) 2015 1500123).

Given these two equilibria, and given the overall constraint of the available driving force from exciton to free charges for a given blend, the scope of the current manuscript can be summarized like this: what is the optimum CT energy to minimize V_{oc} losses while still avoiding EQE losses?

We have included a new Figure 4c and a corresponding discussion, in which we show the general usefulness of our finding: depending how resilient the IQE of a D-A combination is towards low driving forces, the maximum achievable PCE will depend strongly or weakly on the CT energy. This notion is very useful to avoid trial and error in morphology optimization, a tedious procedure with many experimental parameters.

Now the text reads as follows:

“Our model is generally valid for donor-acceptor blends of low driving force, so that an equilibrium LE-CT is formed. As such, it has significant impact for material search and device optimization in OPV. Consider the case of Y6, also boasting high IQE and low voltage losses in some of its blends.³⁸ This NFA is designed to be particularly rigid, thus combining good charge transport properties with long exciton lifetimes. It can be shown that long exciton lifetimes shift the normalized IQE (orange curve in Fig. 4c) to the left: the material can thus keep unity exciton splitting efficiencies for even lower driving forces.³⁶ This means that the maximum of the resulting normalized PCE (green curve in Fig. 4c) will be much broader for Y6 than for O-IDTBR in a given blend, so that for Y6, adjusting the CT energy to an optimum value is less critical for maximizing PCE as it is for O-IDTBR. By predicting whether a dedicated morphology optimization will lead to a PCE gain in a given material combination, application of our model can be used to avoid a tedious trial and error process. In this way, it broadens the range of possible candidate materials by lowering the PCE threshold for consideration.”

Reviewer #3, point 14:

12) The authors state that the equilibria take place between states in the bulk (LE, CS) and at the interface (CT), meaning that there is diffusion/transport of excitons/charges between the states. How is this taken into account in the model? How will it affect equilibration?

Author response:

We thank the referee for highlighting this point. If equilibration is mediated by diffusion, then the question is whether detailed balance can be installed during the lifetimes of the contributing excited states. From the measured EQE values (exceeding 80%), we conclude that the IQE values are above 90%, which means that the interface can be reached by excitons formed almost anywhere inside the bulk. Due to the stochastic nature of diffusion (as distinguished from a drift), it follows that also an LE state produced at the interface via the $LE \Leftrightarrow CT$ equilibrium, can reach any point inside the bulk. We therefore conjecture that detailed balance is indeed established in our experiments, which allows us to take diffusion implicitly into account by introducing a Boltzmann degeneracy factor $g_{LE,CT}$ referring to a ratio of interfacial and bulk states.

We introduced this discussion into the revised Supporting information.

changes: Supporting information, pg. 24, now the text reads as follows:

“In Supplementary Table 3, we summarize the fixed parameters and their provenience. Here, we comment on the degeneration ratios from reference(iv): From the measured EQE values (exceeding 80%), we conclude that the IQE values are above 90%, which means that the interface can be reached by excitons formed almost anywhere inside the bulk. Due to the stochastic nature of diffusion (as distinguished from a drift), it follows that also an LE state produced at the interface via the $LE \Leftrightarrow CT$ equilibrium, can reach any point inside the bulk. We therefore conjecture that detailed balance is indeed established in our experiments, which allows us to take diffusion implicitly into account by introducing a Boltzmann degeneracy factor $g_{LE,CT}$ referring to a ratio of interfacial and bulk states.”

Reviewer #3, point 15:

13) Some of the parameters are taken in un-contacted films (TA, I presume), while others come from device measurements. Whether there is extraction or not will however strongly affect the equilibria that the authors talk about. How will the presence/absence of extraction affect the CS rise that is fixed here from TA?

Author response:

We thank the referee for highlighting this point. We explicitly considered the presence or absence of charge extraction in our multi-objective optimization scheme, by setting the extraction time constant to either infinite (to simulate EL spectra and EL quantum yields) or used it as fitting parameter (to find IQE).

Our method to measure charge separation dynamics relies on the transient Stark effect, which is sensitive to the first steps of charge separation. These occur on a time scale which is 3 orders of magnitude faster than charge extraction. The influence of the presence of an extraction field on these first steps of charge separation should therefore be marginal.

We have included these points into the revised material.

Now the text reads as follows:

“We highlight that the charge separation time of 40 ps has been obtained in a thin film without extraction layers, while EQE was measured at short circuit and hence under an extraction field. As the

charge separation time refers only to the first step of charge separation, we do not expect a significant influence of an extraction field of the charge separation time."

Reviewer #3, point 16:

14) The reason of low EQE with a small LE-CT offset is that (according to e.g. Marcus theory), the exciton dissociation becomes slow (although recent work shows this is not necessarily the case). This is not discussed at all in the manuscript, and the energetics is assumed to only affect the equilibrium between the two states. However, the slow acceptor exciton decay (compared to instantaneous excited donor splitting) points to different exciton splitting times. In general, what do the authors expect to change when the acceptor or donor is initially excited (they mention excess energy in the CT state but do not pursue this further)?

Author response:

We thank the referee for posing this question, which has not been addressed rigorously in the manuscript. If the donor is initially excited, we have a high driving force situation, and hence a non-equilibrium for which Marcus theory has not been formulated; transfer will therefore nearly always be ultrafast as long as it is not diffusion limited (J.Unger et al., J. Phys. Chem. C 2017, 121, 41, 22739–22752) If the driving force is low, then Marcus predicts an activation barrier causing slower transfer depending on the assumed values for the donor-acceptor coupling and the reorganisation energies. For typical values in high-performance blends, one arrives at Marcus transfer times in the tens of ps range. This is much slower than typical relaxation times of "hot" excitons, suggesting that Marcus electron transfer (in the Marcus-Levich-Jortner semiclassical formulation) can adequately describe these situations. Some of us have found in a recently published paper (ref. 36 in main manuscript) that as long as the exciton dissociation rate is higher than 10^{10} s^{-1} , it does not influence the multiobjective modelling. Therefore, we have fixed it to 10^{11} s^{-1} in the model.

We refer to this paper into the revised version.

Now the text reads as follows:

"In the model, we found that $k_{(LE,CT)}$ plays no role as long as it is higher than 10^{10} s^{-1} , a condition safely met in highly efficient D-A blends[Ref 36]; therefore we fixed this value to 10^{11} s^{-1} , so that the remaining number of free parameters is four"

Reviewer #3, point 17:

15) Data with PCBM is initially shown, but this is then not exploited further in the data interpretation. Using the kinetic model (or an alternative one) on this system (with largely different energetics and a clearly distinct CT state) would make the story much stronger.

Author response:

Our model is based on an equilibrium between LE and CT states. If the driving force is as strong as in WF3:PCBM, such an equilibrium will not form, and therefore we cannot describe this system with our model. Moreover, it has been shown that in high driving force systems, ultrafast coherent effects are predominant: the excess energy is efficiently used for coherent long range transfer in a band of states. Such phenomena are however not possible if the driving force is small, and equilibria have to be considered. Therefore, we have focused our modelling on this new class of high-performance blends of low driving force.

REVIEWER COMMENTS

Reviewer #1 (Remarks to the Author):

I appreciate the efforts that the authors made on addressing the comments and improving the manuscript. I am happy with the revisions, but still not sure whether it merits the publication in Nature Communications for the following reason:

During the revision of the manuscript, I notice that some of the same authors published a very nice paper (Nature Energy 2020, DOI: 10.1038/s41560-020-00684-7). I think that the Nature Energy paper significantly affects the novelty of the current manuscript. The materials and the thermodynamic modelling in the current manuscript is the same as what were shown in their Nature Energy paper. What is new here is that the authors try to explain the results from a different angle; however, this 'different angle' is somehow obvious from the results in the Nature Energy paper, especially from Figure 3. What I suggested in my previous report (to fix one component and tune the energetic offset by changing the other component) is exactly what the authors did in their Nature Energy paper.

Reviewer #2 (Remarks to the Author):

Authors have taken into account my concerns and comments and have significantly improved the clarity of the manuscript.
I recommend publication.

Reviewer #4 (Remarks to the Author):

I think this is a very nice paper, and a lovely follow-up to their Nature Energy paper of this year. However, I think its complexity makes it less suitable for Nature Communications: so much key information needed to appreciate the results is still in the SI.

In their rebuttal letter, the authors state that they believe this to be the first experimental evidence of ultrafast charge separation in a low driving force blend but this is not the case. Banerji et al showed this in their Nature Comm paper 2020 (Nature Communications volume 11, Article number: 833 (2020): this paper is not cited here. Another very relevant paper that is not cited is Chow et al. Nature Communications volume 11, Article number: 5617 (2020).

I could not find the exact excitation wavelength used for the TA spectroscopy, just a comment in the methodology that the pump pulses span a certain wavelength range, and that it is primarily the donor that is excited. That being said, is this truly selective excitation? How much acceptor excitation is also occurring? Is there any energy transfer occurring as well? Why was donor excitation employed, when acceptor excitation could achieve complete selectivity? Are the same results acquired with acceptor excitation? I note that radiative/non-radiative decay from the donor - such as energy transfer - is not considered in their model (Fig 3), but it is hard to believe that the charge separation negates ALL of these other processes. As such, can the authors be confident that the acceptor excitons observed at such early times are formed solely via the LE/CT equilibrium? Furthermore, the authors talk about the importance of the HOMO-HOMO offset, but then proceed to do their TA experiments with donor excitation, which proceeds largely via the LUMO-LUMO offset. I think there are still quite a few unjustified assumptions in this paper.

Oh, and I think there is an error in the author affiliations!! The authors should check this!

REVIEWER COMMENTS

Reviewer #1, point 1 (Remarks to the Author):

I appreciate the efforts that the authors made on addressing the comments and improving the manuscript. I am happy with the revisions, but still not sure whether it merits the publication in Nature Communications for the following reason:

During the revision of the manuscript, I notice that some of the same authors published a very nice paper (Nature Energy 2020, DOI:10.1038/s41560-020-00684-7). I think that the Nature Energy paper significantly affects the novelty of the current manuscript. The materials and the thermodynamic modelling in the current manuscript is the same as what were shown in their Nature Energy paper. What is new here is that the authors try to explain the results from a different angle; however, this 'different angle' is somehow obvious from the results in the Nature Energy paper, especially from Figure 3. What I suggested in my previous report (to fix one component and tune the energetic offset by changing the other component) is exactly what the authors did in their Nature Energy paper.

Author response:

We thank the reviewer for their positive comments on the revised manuscript as well as the Nature Energy paper published by some of us. However, we kindly disagree with the reviewer about the novelty of the present manuscript. In fact, the Nature Energy paper demonstrates a Boltzmann equilibrium between excitons and charge transfer states in blends featuring low driving force. In contrast, the present manuscript builds on this finding and goes one step further, dealing with the equilibrium between charge transfer states and charge separated states, i.e. free charges. In particular:

- 1) We present extensive transient absorption data with high temporal resolution, which reveals kinetic information beyond what we obtained in the Nature Energy paper.
- 2) We combine this data with extensive device and quantum chemical data to characterize the charge separation process, which takes place in 40 ps.
- 3) We demonstrate that charge separation in this blend must be an energetically downhill process.
- 4) We show that, for any low driving force blend, there exists an optimal energy for interfacial charge transfer states to maximize device performance.

All these results vastly expand the findings of the Nature Energy paper.

Point number 3, in particular, supports recent conclusions reported by Prof. Neher et al. (Advanced Materials 2020, **32**, 1906763) about “activationless charge separation” by evidence in the time domain, and challenges a very recent paper (Chow et al. Nature Communications volume 11, Article number: 5617 (2020)) claiming endothermic charge separation. In our opinion, our manuscript is therefore an important and novel contribution to a lively discussion about the role of charge separation at low driving forces in an emerging class of materials which are currently revolutionizing the field of organic photovoltaics.

Reviewer #2 (Remarks to the Author):

*Authors have taken into account my concerns and comments and have significantly improved the clarity of the manuscript.
I recommend publication.*

Author response:

We thank the referee for the positive evaluation of the manuscript and for recommending publication in the present form.

Reviewer #4, point 1 (Remarks to the Author):

I think this is a very nice paper, and a lovely follow-up to their Nature Energy paper of this year. However, I think its complexity makes it less suitable for Nature Communications: so much key information needed to appreciate the results is still in the SI.

Author response:

We thank the referee for their positive evaluation of the manuscript and for their constructive comments. We understand the criticism about the complexity of our manuscript. We are however convinced that, while our analysis is complex as it needs to take into account inputs from different sources (ultrafast transient absorption spectroscopy, quantum chemical calculations, device characterization), the results are quite straightforward to understand and of broad interest for the community. For this reason, we have left in the main text only the conclusions of our analysis and put the detailed description of our approaches in the Supplementary Information, to enable interested readers to reproduce them. In our revised version, to further reduce the complexity, we have clarified the wording in the Supplementary Information, part B, to explain how we obtain the photoexcitation dynamics from the characteristic dynamics of the singular value decomposition. Previously, it was necessary to read through the mathematical reasoning in the Appendix to understand this, which was tedious. We hope that this modification facilitates the understanding of our method for those readers not interested in its technical details.

In addition, we have found an inconsistency between the nomenclature of charge dynamics in the Supplementary information, part 2, and the Appendix of the Supplementary information ($n_i(t)$ and $s_m(t)$, respectively). We apologize for this mistake which made it hard for the reader to follow our methodology. This has been corrected in the revised Supplementary Information.

Changes:

Revised Supplementary Information, part 2 (page 9).

Reviewer 4, point 2:

In their rebuttal letter, the authors state that they believe this to be the first experimental evidence of ultrafast charge separation in a low driving force blend but this is not the case.

Banerji et al showed this in their Nature Comm paper 2020 (Nature Communications volume 11, Article number: 833 (2020): this paper is not cited here. Another very relevant paper that is not cited is Chow et al. Nature Communications volume 11, Article number: 5617 (2020).

Author response:

We thank the referee for highlighting these recent publications, which we have now added to the revised version of the manuscript; they indeed increase the impact of our discussion part.

The paper by the Banerji group is focusing on exciton dissociation, that is, formation of the charge transfer complex, while our manuscript is concerned with the separation of charge transfer states

into free charges. Although these authors use the term “charges”, they explicitly show in their fig. 2C that they are referring to charge transfer states. They find exciton dissociation times below 1 ps, fully supporting our picture that in low offset blends, charge generation dynamics is not limited by exciton dissociation.

The paper by Chow et. al indeed demonstrates charge separation in the 100 ps time domain. However, according to the recent publication by the Banerji group, the slow exciton dissociation in the tens of picoseconds timescale observed in Chow et al. may be diffusion limited, thus hiding the real charge separation kinetics. Additionally, the choice of the 800-nm wavelength for the push pulse may not be optimal, as this pulse will also dump excitons to the ground state, in addition to the desired effect of re-exciting CT states.

In summary, the two papers recently published in Nature Communications are important contributions to the field of organic photovoltaics with non-fullerene acceptors and testify the broad interest of the topic. However, we insist that our paper is the first one quantifying charge separation kinetics in blends of low driving force. This claim is however not part of our key claims in the manuscript itself.

In the revised version, we include these two works in the discussion. We highlight that our experimental evidence does not support the picture of the Chow group, finding endothermic charge separation, but rather points towards exothermic charge separation, which supports recent reports about “activationless charge separation”, possible in intercalating interfaces [Cha et al. Advanced Energy Materials 2019, 9, 1901254] or caused by quadrupolar moments [Advanced Materials 2020, 32, 1906763 and Nature Materials 2020, 10.1038/s41563-020-00835-x].

Changes:

Revised manuscript, page 10 and page 14.

Reviewer 4, point 3:

I could not find the exact excitation wavelength used for the TA spectroscopy, just a comment in the methodology that the pump pulses span a certain wavelength range, and that it is primarily the donor that is excited. That being said, is this truly selective excitation? How much acceptor excitation is also occurring?

Author response:

We apologize with the referee for missing this important information. In the revised manuscript, we included the spectrum of the pump pulse. From the spectral shape of the pump pulse, it becomes clear that excitation is not truly selective; it is mainly the donor polymer that is excited. From considering the pump pulse and the absorption spectrum, we estimate that our pump pulse generates 70% donor excitons and 30% acceptor excitons, see new Supplementary Figure 25. However, please note that the maximum of the green curve in Fig. 2F of the manuscript does not correspond to the pump-induced acceptor exciton concentration, because equilibration between CT and LE states is expected in the sub-picosecond time domain.

In the revised version, we included this discussion, as follows:

“The samples are excited by a 10-fs pulse, mainly in resonance with WF3 excitons (shaded grey in Figure 2c), and probed with a broadband continuum covering the 1.3-2.2 eV photon energy range. We choose broadband excitation because it yields to a much better time resolution compared to narrowband excitation.”

Changes:

Pump pulse introduced in Fig.2; discussion on page 7, new Supplementary Figure 25.

Reviewer 4, point 4:

Is there any energy transfer occurring as well?

Author response:

We thank the referee for highlighting this point. Energy transfer from donor towards acceptor excitons is doubtlessly expected to occur, because the emission spectrum of the donor is partly overlapping with the absorption spectrum of the acceptor. However, typical energy transfer times are in the few picoseconds range (see, e.g. the paper from the Banerji group mentioned above, Fig. 2C, purple arrow). Therefore, energy transfer is completely outperformed by ultrafast exciton dissociation.

We can demonstrate this by comparing the transient absorption spectra in Fig. 2c and 2d: if energy transfer were dominant, then the TA spectrum of the blend (black line in Fig. 2d) should look like the TA spectrum of acceptor excitons (green line in Fig. 2c). In contrast, our spectral decomposition shows only a minor contribution of acceptor excitons, which approximately agrees with the amount of resonantly created acceptor excitons (approximately 30%, see new Supplementary Figure 25). Therefore, we can experimentally exclude a significant contribution of energy transfer besides charge transfer.

Changes: we added this point to revised manuscript, pg.10, as follows:

“The spectral decomposition, as shown in Fig. 3d, yields a small initial population of acceptor excitons (dark green spectrum). The relative strength of this contribution approximately agrees with the expected amount of resonantly created acceptor excitons (about 30%), obtained by multiplication of the pump pulse spectral density with the pure donor and acceptor absorption spectra (Supplementary information, Part 10). Therefore, we can associate the initial population of O-IDTBR excitons with resonant creation, showing that ultrafast energy transfer from donor to acceptor excitons does not play a significant role in our samples.”

In addition, please check the new supplementary Figure25.

Reviewer 4, point 5:

Why was donor excitation employed, when acceptor excitation could achieve complete selectivity?

Author response:

We have chosen a broadband pump pulse mainly in resonance with donor excitons because it presented the best compromise between time resolution and energy selectivity. Choosing a pump pulse which is exclusively in resonance with the acceptor would have required a much narrower bandwidth (from new Supplementary Figure S25, it can be seen that there is only a very narrow spectral range from 1.6 to 1.7 eV where exclusively o-IDTBR absorbs) and a concomitant reduced temporal resolution.

Changes:

We included this point into the revised version, page 7, as follows: “.

We choose broadband excitation because it yields to a much better time resolution compared to narrowband excitation.”

Referee 4, point 6:

Are the same results acquired with acceptor excitation?

Author response:

We apologize for not mentioning the availability of these experiments in the main manuscript. We have performed a TA study using narrowband pump pulses tuned to either the donor or the acceptor phase. The results are given in the Supplementary Information, section 6. For predominant donor excitation, we observe similar TA spectra as for broadband excitation. For selective acceptor excitation, we observe the influence of exciton diffusion, which has been found also by the Banerji group and explained by a shorter wavefunction localization in the small molecule system compared to polymers (page 4 in the paper of the Banerji group, right, citing her paper from 2013).

Changes:

Inclusion of this discussion of page 12 in the revised manuscript as follows:

“We have performed a comparative study using narrowband excitation exclusively in resonance with the acceptor or the donor exciton (Supplementary material, Part 6). These studies are qualitatively in agreement with the results of Fig. 2 albeit with a much longer instrumental response function, due to the narrowband pump pulses. We also observed a possible diffusion limitation of hole transfer after resonantly exciting the acceptor exciton; this has been found recently and ascribed to a lower wavefunction extension in the small molecule NFA system as compared to the donor polymer.³² For this reason, we did not use the data from the narrowband experiment for kinetic modelling.”

Referee 4, point 7:

I note that radiative/non-radiative decay from the donor - such as energy transfer - is not considered in their model (Fig 3), but it is hard to believe that the charge separation negates ALL of these other processes. As such, can the authors be confident that the acceptor excitons observed at such early times are formed solely via the LE/CT equilibrium?

Author response:

One highlight of our experimental results is the occurrence of sharp derivative-like features, identifying charge transfer states, on a time scale as early as 50 fs, owing to the higher time resolution of our experimental setup in comparison to the one usually employed in other studies, such as e.g. by Banerji's group. This extremely rapid exciton dissociation time largely outperforms any of the other non-radiative decay processes starting from the S1 excited state of the donor, namely internal conversion of S1 to the ground state S0 (few hundred picoseconds) and energy transfer towards acceptor excitons (few picoseconds). Therefore, it is not required to include these processes into the kinetic scheme.

Changes:

Inclusion of this discussion in the revised manuscript, pg.11, as follows:

“An interesting point is the decay of acceptor excitons on the same time scale as the build-up of the CS state (compare green and black curves in Fig. 2f). This is a direct consequence of the LE \rightleftharpoons CT equilibrium: the femtosecond pump pulse predominantly generates excitons in the donor phase. Due to the high driving force for exciton splitting, they convert instantaneously into CT states. However, acceptor LE states are accessible from CT states, and therefore an equilibration must occur.”

Reviewer 4, point 8:

Furthermore, the authors talk about the importance of the HOMO-HOMO offset, but then proceed to do their TA experiments with donor excitation, which proceeds largely via the LUMO-LUMO offset.

Author response:

We have chosen a broadband pump pulse mainly in resonance with donor excitation to improve temporal resolution beyond the typical literature standard. Our Fig. 2D shows that CT states are formed within 50 fs; our Fig. 2F shows that they are in equilibrium with acceptor excitons and form charge separated states. Therefore, even though we initially excite donor excitons, after 50 fs we are only observing photoexcited states whose photophysics is governed by the HOMO-HOMO offset.

Changes:

We highlight this point in the discussion of Fig. 2 in the revised manuscript (page 10) as follows:

“Since the formation of CT states from donor excitons outperforms all possible deactivation channels, it follows that the photoexcitation dynamics of our systems only depends on the HOMO-HOMO offset; the LUMO-LUMO offset does not contribute quantitatively as long as it is above a certain threshold value allowing the observed ultrafast transfer.³³”.

Reviewer 4, point 9:

I think there are still quite a few unjustified assumptions in this paper.

Author response:

In our opinion, we have supported our key claims adequately by literature results or by our own experimental data, where applicable; we have done so also in this Response Letter. Therefore, we beg to disagree on this comment by reviewer 4, that there be unjustified assumptions. If the reviewer could provide evidence for this claim, we would be happy to address the points and justify our adopted hypotheses. We hope that the unnamed unjustified assumptions mentioned here were part of the previous comments and have been clarified in the revised manuscript.

Reviewer 4, point 10:

Oh, and I think there is an error in the author affiliations!! The authors should check this!

Author response:

We thank the reviewer for spotting this error; the authors' affiliations are now correct.

REVIEWERS' COMMENTS

Reviewer #1 (Remarks to the Author):

After reading the responses and changes to my comments also to those of Referee 4, I think that the manuscript is now acceptable by Nature Communications.

Reviewer #2 (Remarks to the Author):

I feel the paper is of sufficient novelty and quality to be published in nature communications, once the issues raised by the other referees are resolved.

Reviewer #4 (Remarks to the Author):

Thank you for addressing my comments so fully, I am now happy to recommend acceptance of this manuscript. In particular I think clarifying SVD part in the supporting info has helped a lot. Also, I did not truly appreciate the broadband nature of the excitation in the first version I read. I think this has indeed been highlighted more strongly, which is great... but I would go one step further. Rather than the casual comment "we choose broadband excitation because it yields a much better time resolution compared to narrowband excitation", point out more emphatically that it leads to much better time resolution than "standard" ultrafast TAS and this gives you the advantage of accessing times prior to most other photophysical processes, etc, etc. Since this is one of your main points of novelty/impact (and enables greater confidence in your model), it is worth even greater emphasis.

REVIEWER COMMENTS

Reviewer #1 (Remarks to the Author):

After reading the responses and changes to my comments also to those of Referee 4, I think that the manuscript is now acceptable by Nature Communications.

Author response: We thank the reviewer for their positive comment.

Reviewer #2 (Remarks to the Author):

I feel the paper is of sufficient novelty and quality to be published in nature communications, once the issues raised by the other referees are resolved.

Author response: We thank the reviewer for their positive comment.

Reviewer #4 (Remarks to the Author):

Thank you for addressing my comments so fully, I am now happy to recommend acceptance of this manuscript. In particular I think clarifying SVD part in the supporting info has helped a lot. Also, I did not truly appreciate the broadband nature of the excitation in the first version I read. I think this has indeed been highlighted more strongly, which is great.... but I would go one step further. Rather than the casual comment "we choose broadband excitation because it yields a much better time resolution compared to narrowband excitation", point out more emphatically that it leads to much better time resolution than "standard" ultrafast TAS and this gives you the advantage of accessing times prior to most other photophysical processes, etc, etc. Since this is one of your main points of novelty/impact (and enables greater confidence in your model), it is worth even greater emphasis.

Author response: We thank the reviewer for their positive comment. We have revised the sentence in the manuscript accordingly. Now reads as:

“We choose broadband excitation because the resulting 10-fs pulses yield a much better time resolution compared to standard TA measurements with time resolution of the order of 100 to 200 fs, allowing us to observe photophysical processes from their very onset.”